# Discovery of Hierarchy in Embedding Space

## Abstract

Existing learning models partition the generated representations using linear hyperplanes which form well defined groups of similar embeddings that can be uniquely mapped to a particular class. However, in practical applications, the embedding space do not form distinct boundaries to segregate the clusters. Moreover, the structure of the latent space remains obscure. As learned representations are frequently reused to reduce the inference time, it is important to analyse how semantically related classes interact among themselves in the latent space. We have proposed a cluster growing algorithm that minimises the inclusion of other classes in the embedding space to form clusters of similar representations. These clusters are overlapping to denote ambiguous embeddings that cannot be mapped to a particular class with high confidence. Later, we construct relation trees to evaluate our method with the *WordNet* hierarchy using phylogenetic tree comparison methods.

## 1 Introduction

In modern computer vision, utilising the learned high-dimensional embeddings to reduce the inference time is ubiquitous. Conventional models aim at grouping images based on semantic relations which are later assigned to classes. The assessment of similarity among the images are achieved using simple distance measures. Thus, specific geometry of the embedding space is implied by the operations at the end of deep models. For example, classification networks (Krizhevsky et al., 2017) use linear operators to map embeddings of the penultimate layer to its respective classes. This indicate that the embeddings learned by the model lie in the Euclidean space. Each pair of classes in the latent space are separated by Euclidean hyperplanes. Hence, Euclidean distances are often used to perform face recognition (Wen et al., 2016), one-shot learning (Snell et al., 2017) or image retrieval (Wu et al., 2017). Some methods also use spherical embeddings by applying a spherical projection operator for computing the embeddings at the end of the network. However, the partitioning of embeddings to form well defined groups by the hyperplane is enforced by the model to assign each representation to unique classes. In Fig. 1, scatter plots of the generated feature representations from different models are shown as points in a 3-dimensional space.

Hyperbolic spaces have negative curvature unlike Euclidean and spherical spaces which have zero and positive curvatures, respectively. These have profound effect on the nature of embeddings that the current methods can learn (Khrulkov et al., 2020). It has been observed that embeddings in hyperbolic space, perform significantly better that those in Euclidean space. Thus, it is yet to completely understand the nature and structure of the latent space under various circumstances. Although generating better visual embeddings has been an active area of research today, most of these works overlook the interaction among the classes while learning class representations.

We propose to analyse the structure of the embeddings formed without enforcing any particular criterion for training. The generated representations are assumed to be points in high dimensional space on which we apply a cluster growing algorithm to group them based on their similarity. These class associated embedding clusters (CAEC) are not well partitioned and have overlaps to encode embeddings which are ambiguous and can be used to represent more than one classes. Such highly related visual features are often observed among closely related classes. Thus, using our cluster growing algorithm, we have been able to capture such non-discriminative class embeddings. The proposed cluster growing technique maximises grouping of same class embeddings and minimises the inclusion of those of other classes. We compare the results observed

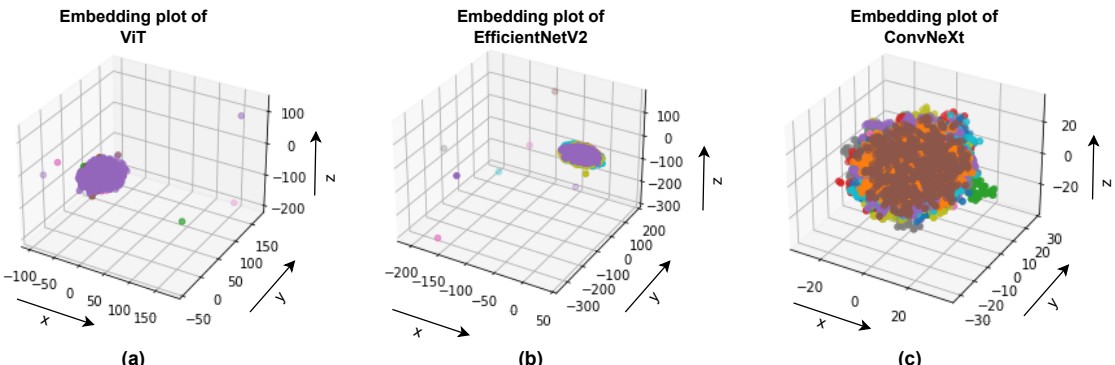

Figure 1: Scatter plots of the embeddings generated from three different models (a) Vision Transformer, (b) EfficientNetV2 and (c) ConvNeXt, in 3D Euclidean space. The number of classes here are 10 and the total number of data points are 50,000.

for the representations generated from three different types of models, namely, Vision transformer (*ViT*) (Dosovitskiy et al., 2021), convolutional neural network (*EfficientNetV2*) (Tan & Le, 2021) and ConvNeXt (Liu et al., 2022). Even though the embedding space is Euclidean, we observe marked differences in the quality of clusters when different types of distance measures are used while clustering. A relation tree is constructed by applying the unweighted pair group method with arithmetic mean (UPGMA) (Dawyndt et al., 2006) on cluster centroids to hierarchically visualise the semantic relations among the classes. This tree is compared with the *WordNet* (Miller, 1995) ontology using phylogenetic tree comparison methods to evaluate our proposed algorithm. We summarise our main contributions as follows:

- Forming class associated embedding clusters (CAEC) using our proposed cluster growing technique.

- Analysing the structure of the clusters and proposing a formal representation to denote each cluster.

- Evaluating the quality of the clusters and comparing the hierarchical relation tree formed for evaluation.

Our proposed method has been able to form good quality clusters along with capturing the semantic relationship among the classes.

## 2 Recent Works

Most of the recent works focus on generating better feature embeddings to enhance the performance of the model on various tasks, such as, image retrieval, classification, recognition and segmentation. These methods use both supervised and self-supervised learning to produce improved quality representations. However, very few work has been done on interpreting the structure and relationship among the representations in the latent space.

### 2.1 Learning feature embeddings

In the field of computer vision, most researchers aim at improving the quality of features through both supervised and contrastive methods. However, it has been observed that there exist a disparity among the representations generated for various tasks, such as, classification and segmentation. Thus, feature embeddings are generated primarily based on the application, although lately Oquab et al. (2023) have proposed on producing general-purpose embeddings. Moreover, an abundance of un-annotated data has shifted the focus to self-training methods, where the embedding quality is improved by training a large set of unlabeled data using a small set of annotations. We have divided the recent works on feature learning into supervised and self-supervised methods.

### 2.1.1 Supervised methods

The types of features learnt depend solely on the goals of the vision task. Image features obtained using convolutional neural networks have achieved state-of-the-art performance in classification task. Learning these embeddings aim at determining well defined hyperplanes in the feature space. However, in image retrieval, the similarity among the classes are exploited to minimise and maximise the intra-class and inter-class distances in the feature representations (Liu et al., 2017). In Zhang et al. (2016), a method has been developed to combine multiple losses to embed more information in the features. The authors incorporate additional label information, such as hierarchy or shared attributes, to the representations to form the final embeddings. Kan et al. (2019) demonstrate that fusing deep features with handcrafted features result in better representations if the two types of features are complementary.

Multi-modal feature extraction using the structural and semantic correspondences between the visual and textural features is proposed by Ge et al. (2021). The two feature embeddings were learnt jointly using a common context-free referral tree. *DeepVoxels* (Sitzmann et al., 2019) give latent representations to input images of a scene instead of constructing its geometry. These representations can be directly used to generate 3D scenes without utilising the initial set of input images. Euclidean and spherical embeddings currently dominate computer vision tasks in a way that the degree of similarity to determine class memberships are determined using linear hyperplanes. Khrulkov et al. (2020) demonstrate the benefits of using hyperbolic image embeddings. Initially they have used hyperbolic neural networks to generate the embeddings. Later they evaluated the hyperbolicity of the data set which enables them to estimate the radius of Poincaré disk for an embedding of a specific data set.

### 2.1.2 Self-Supervised methods

Recently, researchers have been dwelling on self-supervised learning to formulate discriminative approaches to learn feature representations that can be used in downstream tasks. These networks have either convolutional backbones (Bardes et al., 2022a; Tomasev et al.) or Vision transformer (Li et al., 2022; Zhou et al., 2022). Current approaches include selecting pairs of views of the same image and learning invariant features using a joint embeddings architecture (Misra & Maaten, 2020; He et al., 2020). Misra & Maaten (2020) eliminate the irrelevant part of information on position and colour to produce better classification results on benchmark data sets, while He et al. (2020) use momentum contrast to learn visual features in an unsupervised fashion. While the above methods strive to outperform state-of-the-art results in classification tasks by improving on global features, Yang et al. (2022) and Hénaff et al. (2022) extract local features and form embeddings that perform well in semantic segmentation task. Bardes et al. (2022b) have used position-based and feature-based matching to devise the local criterion that outperform most of the related works on segmentation without compromising on global features which are learnt using variance and covariance loss.

A considerable section of self-supervised learning apply instance classification to learn discriminative features. In instance classification, each image is treated as a different class and the model is trained by discriminating them. However, it is difficult to generate such embeddings when the number of images increases (Dosovitskiy et al., 2016). Grill et al. (2020) train features by matching them to the representations generated by a momentum encoder. This unsupervised technique learn features without discriminating between images.

## 2.2 Manifold learning method

High dimensional data are often mapped to lower dimensions for better representations. The mapping function can be provided or learned by the model. Manifold learning algorithms are used to find low dimensional parameterization of high-dimensional data (Zhang et al., 2011) which enables the understanding of the intrinsic structures, feature analysis and visualization. Zhang et al. (2011) proposes an adaptive neighbourhood selection and bias reduction method in local tangent space alignment (LTSA) (Zhang & Zha, 2003) to form better approximations of high dimensional embeddings. Manifold learning is beneficial compared to classical decomposition methods such as principal component analysis (PCA) as it preserves distance metrics locally with different manifold mapping (Van der Maaten & Hinton, 2008) and beneficial at retaining the intrinsic local geometrical structure (Li et al., 2017) in low dimensional space (Han et al., 2022) such as in

t-distributed stochastic neighbor embedding (t-SNE) (Van der Maaten & Hinton, 2008; Li et al., 2017) and uniform manifold approximation and projection (UMAP) (Becht et al., 2019).

### 2.3 Interpreting latent representations

In deep learning, model interpret data in high dimensional space as feature maps which are later encoded as feature vectors, commonly referred to as latent representations. These representations retain only relevant information from the initial data which can later be used for downstream tasks (Bengio et al., 2013). However, the acquired information cannot be interpreted easily as the representations are often unstructured (Klys et al., 2018). In absence of any regularisation in the latent space, these representations demonstrate obscure structure (Mathieu et al., 2019). Higgins et al. (2017) proposed to constraint the latent space capacity resulting in learning only salient features of the data to establish more explicable representation.

Many authors have attempted to decipher the structure of the latent space empirically. Cristovao et al. (2020) tried to generate images at different resolutions using latent representations which did not perform well. As the generated images did not resemble the structure of the original image, they assumed that the learning representations were not constrained under the latent space and have certain degrees of freedom. Based on empirical results, they have also claimed a probable explanation that the structure of the latent space was not suitable for interpolation. Hence, they designed a network to enforce an appropriate structure to the latent space that would enable them to use the interpolation method for generating in-between images.

In our proposed method, empirically we try to interpret the interaction among the classes in embedding space using a cluster growing algorithm. We assume that there exist inherent groupings and semantic relations among the classes present in a data set. The representations formed by the encoder network for every class are not well defined or discriminative to that class due to these interactions. Therefore, the structure of the latent space may embed the semantic relationship among the classes. We analyse the change in the structure of these representations when different types of models are used. Moreover, the cluster formation vary when distance measures are modified. The novelty of this method lies in the cluster growing algorithm to form overlapping clusters instead of well partitioned groups to interpret the relationship among the classes. We have also been able to construct a relation tree using phylogenetic tree construction method and compare the tree with the *WordNet* hierarchy to evaluate our method. We also analyse the structure of the clusters formed by proposing a novel mathematical representation.

## 3 Proposed Method

We generate representations from three different models and apply our cluster growing technique to analyse the structure of the embedding space. The quality of the clusters and the interaction among the classes vary for different types of models. Later, we form relation trees to define the existing similarity among the classes hierarchically.

### 3.1 Generating Image Embeddings

Image embeddings are generated using three types of models, Vision transformer (*ViT*), *EfficientNetV2* and *ConvNeXt*. We analyse the quality of embeddings generated by these models using the proposed cluster growing algorithm.

#### 3.1.1 Vision Transformer

Transformers (Vaswani et al., 2017) have become de-facto standard for natural language processing (NLP) tasks due to their computational efficiency and model scalability. The standard approach is to train transformer models on large text corpus and fine-tune on smaller datasets. The scalability of the model has made it possible to train unprecedented amount of data without overfitting. Vision transformer (ViT) (Dosovitskiy et al., 2021) replicate standard transformers with minor modifications to utilise the benefits of self-attention based models on images for efficient implementation.

The encoder block of ViT comprises of alternate layers of self-attention and linear layers. Every block is preceded and followed by Layernorm and Residual connections, respectively. The activation function used in each block is *GELU* (Nguyen et al., 2021). Standard transformers take sequence of 1D token embeddings with a fixed latent vector of size, say $D$. Therefore, to represent image in a similar manner, a sequence of 2D patches along with their positional embeddings are sent as input vectors. These patches undergo linear projection and mapped to $D$ dimension. We flatten the 2D representations of the final encoder block to form vectors for each image.

### 3.1.2 Convolutional Neural Networks

*EfficientNets* (Tan & Le, 2019) are a family of light weight *ConvNet* models that are both parameter efficient and produce state-of-the-art classification results. The baseline model is fixed and designed using neural architecture search which is later scaled up for better accuracy based on the availability of the resources, to form a family of models. Compound scaling is used to uniformly scale up all the dimensions like depth, width and resolution. The architecture uses inverted mobile bottleneck, MBConv, as the main building block and also add squeeze-and-excitation optimization to it.

We have used the *EfficientNetV2* (Tan & Le, 2021) model, which is an improved version of *EfficientNets*. This model has better parameter efficiency and training speed compared to the previous models. The models have been developed using a combination of training-aware neural architecture search and compound scaling to optimise the training speed and parameter efficiency. The embeddings generated from the final block is considered for our analysis.

### 3.1.3 ConvNeXt

Hierarchical transformers such as Swin transformers have introduced several aspects of *ConvNets* to make transformers a generic backbone to vision related tasks. It is equipped with inherent inductive biases present in convolutions, although its training setup and architecture remain significantly different from that of *ConvNets*. *ConvNeXt* (Liu et al., 2022) bridges the gap between transformers and convolutional neural networks to form a modified architecture which is aligned with the hierarchical ViT (Swin transformer) without using any attention based modules.

*ResNet-50* forms the baseline model with training setup similar to *ViT*. This gives better results than the original model. Modifications to the architecture is divided into five parts: 1) macro design which includes a "patchify" (non-overlapping convolution) layer, 2) *ResNeXt* module that replaces convolutions with depth-wise convolutions, 3) inverted bottleneck, 4) large kernel size and 5) layer wise micro designs which reduce the number of activation and norms, and replace *ReLU* and Batch-normalisation with *GELU* and Layernorm, respectively. The output of the final convolutional layer is taken as the feature embeddings to conduct our experiments.

### 3.2 Forming Clusters of Embeddings

The embeddings generated for each of the images from a trained model can be visualised as a data point in high dimensional real space, $\mathbb{R}^n$. After the model is fine-tuned, these embeddings form natural groups of similar representations. The details of the experimental setup are presented in Section 4. Ideally, these groups should constitute of a single class. However, due to interaction among semantically related classes and misclassification while training, these clusters contain embeddings of other classes as well.

Let $\mathbb{C}$ be the number of classes present in the data set. Therefore, $e_{ij}$ denote the embedding of the $i^{th}$ sample belonging to the $j^{th}$ class, where, $j \in \mathbb{C}$. We indicate each of the classes in the high dimensional space using $n - d$ centroids, $G_j$ given by:

$$G_j = \frac{\sum_{i=1}^{n} e_{ij}}{n} \tag{1}$$

where, $n$ is the number of samples present in each class. In a completely balanced data set, $n$ will be constant for all the classes. If we visualise the embeddings in real space, we detect formation of natural

clusters indicating similar embeddings. Each cluster, $X_k, k \in \mathbb{C}$ comprises of representations of primarily one particular class, $e_{ij}, j = k, j, k \in \mathbb{C}$ and some representations of other classes, $e_{ij}, j \neq k, j, k \in \mathbb{C}$.

Assuming $G_j$ to be the centre of cluster $X_k, k = j$, we find the distance of the nearest embedding from the centre. This distance signifies the starting radius, $r$, of the cluster. With every unit increase of $r$, we encounter two types of sets, $Y_r^k$ and $Z_r^k$. These sets can be defined as follows:

$$Y_r^k = \{e_{ij} : |e_{ij} - G_j| \leq r, j = k, j, k \in \mathbb{C}, i \in \mathbb{N}\}$$
$$Z_r^k = \{e_{ij} : |e_{ij} - G_j| \leq r, j \neq k, j, k \in \mathbb{C}, i \in \mathbb{N}\} \tag{2}$$

Thus, set $Y_r^k$ consists of all the embeddings that belong to class $j$ within the radius $r$, while $Z_r^k$ comprises of all other classes within the same radius. Here we have assumed that $j$ is the predominating class in cluster $X_k$.

We define the cluster growing technique using $Y_r^k$ and $Z_r^k$ by restricting the number and types of embeddings that can be included in these two sets. Our main idea is to increase the density of same class embeddings, and minimise the interaction among other classes by some constraints. We first characterise the bounding radius, $r_b$, that limits the inclusion of $e_{ij}$ in $Z_r^k$ by a fraction of $\gamma$ to the total number of embeddings in $X_k$. The criterion for $r_b$ can be given as:

$$r_b = r \quad \text{such that} \quad \begin{cases} |Z_r^k| \leq \gamma |X_k|, & r \leq r_b \\ |Z_r^k| > \gamma |X_k|, & r > r_b \end{cases} \tag{3}$$

where, $|X_k| = |Y_r^k| + |Z_r^k|$. In our experiments, we have considered $\gamma = 0.3$. Eqn. 3 gives the criterion for the upper bound of the radius which restricts the inclusion of other classes. To maximize the purity of the cluster, we track the number of newly added embeddings between $r + 1$ and $r$ for $r + 1 \leq r_b$. The unit distance for which the increase is maximum, we consider that as the boundary of our cluster. The final radius, $r_f$ is defined as follows:

$$r_f = argmax_{r+1 \leq r_b} |Y_{r+1}^k| - |Y_r^k| \tag{4}$$

Hence, the final cluster is the union of both the sets given by $X_k = Y_{r_f}^k \cup Z_{r_f}^k$. Fig. 2 depicts the cluster growing algorithm using five classes.

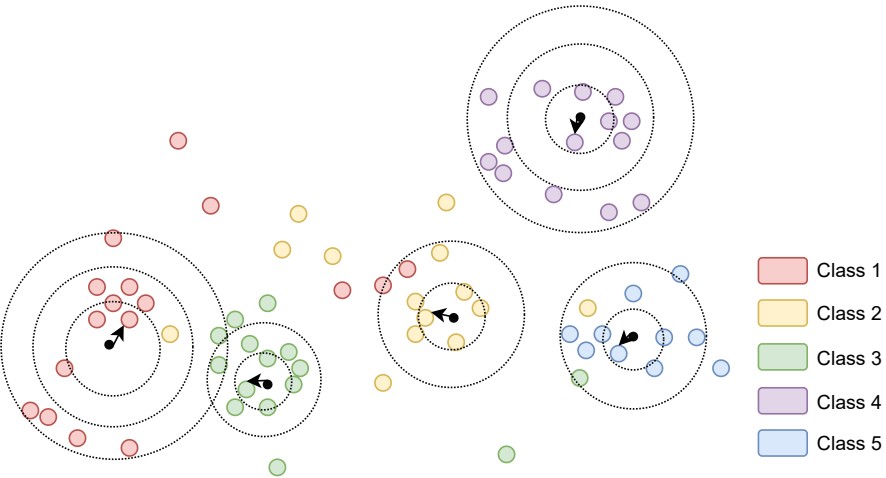

Figure 2: Cluster growing algorithm using five classes.

### 3.2.1 Properties of cluster growing technique

Let $r_i$ be the radius of the nearest target embedding from the centroid, $G_n$ of cluster $c_n$, therefore, the increase in the number of target class embeddings will be given by $|Y_{r+1}^n| - |Y_r^n|$. Similarly, the increase

in the non-target label embeddings is given by $|Z_{r+1}^n| - |Z_r^n|$. Fig. 3 plots the graph of $|Y_{r+1}^n| - |Y_r^n|$ and $|Z_{r+1}^n| - |Z_r^n|$.

**Statement 1:** We observe that all the points in a cluster can be contained within a hypersphere in the

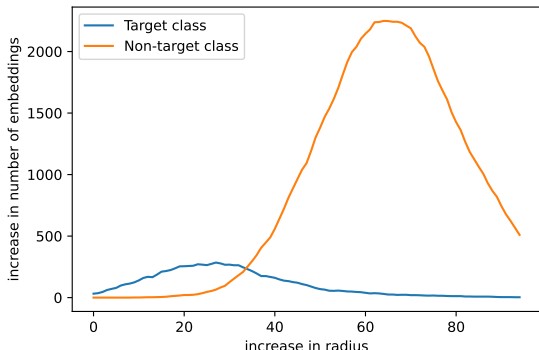

Figure 3: Increase in the number of target and non-target classes with increase in unit radius.

latent space. The embeddings are not present volumetrically in the hypersphere, but rather in a manifold structure. The volume of space covering the centroid, $G$, of each cluster is empty till the first embedding is found at radius $r_{min}$. As the radius is increased by unit measure, new embeddings get included. The last embedding of that cluster form the boundary of the hypersphere. By empirically observing the distributions of number of points with the increasing distance for all the datasets and models under our study, we propose the following hypothesis.

**Hypothesis 1:** The distribution representing the number of embeddings occurring per unit radius, $|Y_{r+1}^n| - |Y_r^n|$, peaks at a particular radius and then starts decreasing.

Though we do not have any theoretical proof of the above hypothesis, intuitively we may consider that for a class in the embedding space the instances lie in a bounded volume. The surface, $\phi_s$, passing through the last point, $e_s$, enclosing all the embeddings within the hypersphere, form the boundary of the embedding space for the particular data set in concern. The centroid, $G_n$, of cluster $c_n$ is present at the centre of the hypersphere representing cluster $c_n$. $\phi_r$ is the surface passing through the nearest embedding to $G_n$. Thus, the density of representations lying on $\phi_r$ is low. If $e_f^n$ be the last embedding of cluster $c_n$, the density of representations lying on surface $\phi_f$ passing through $e_f^n$ will also be low. Therefore, maximum embeddings will lie on the surfaces between $\phi_r$ and $\phi_f$. Hence, we can say that number of embeddings occurring per unit radius peak at a particular radius after which it starts decreasing.

We state in Hypothesis 1 that the distribution followed by the occurrences of embeddings decrease after reaching a peak at a particular radius. This distribution may have one or more peaks. Empirically, we have found that there exists only one such peak per cluster. Thus, we will now identify the nature of this distribution.

**Hypothesis 2:** The function $f(x)$ indicating the increase in the number of embeddings per unit radius, $|Y_{r+1}^n| - |Y_r^n|$, may follow Poisson or Gaussian distribution.

We empirically show that the function $f(x)$ may follow Poisson or Gaussian distribution by applying the chi-squared goodness of fit test. The details of the experimental setup is present in Appendix A. We define the null hypothesis as:

$H_0$ : $f(x)$ follows a given (for example, Poisson, Gaussian, etc.) distribution.

$H_1$ : $f(x)$ does not follow a given (for example, Poisson, Gaussian, etc.) distribution.

The observed number of counts are present using our cluster growing technique. We generate the expected number of counts using Poisson distribution with the same mean. We fix the threshold for $p$-value at 0.05. Thus, if $p$-value$< 0.05$, then we reject the null hypothesis and claim that $f(x)$ do not follow Poisson distribution. Similar experiment is conducted using Gaussian distribution. Table 1 show the chi-squared test result for two distributions.

Table 1: $p$-value observed for the chi-squared.

| Distribution | No. of observations | statistic | $p$-value |
|---|---|---|---|
| Poisson | 22 | 24.47386 | 0.270669 |
| Gaussian | 22 | 0.52469 | 0.989999 |

From Table 1, we observe that the $p$-value$> 0.05$. Hence, we cannot discard the possibility that $f(x)$ follows either Poisson or Gaussian distribution. Empirically we have observed that the $p$-value$> 0.05$ in most cases for Poisson distribution while few clusters may follow Gaussian distribution.

From Hypothesis 2, we may represent $f(x)$ using a Poisson distribution. Given the number of embeddings, we find the probability that it occurs in a given interval of radius.

Using Hypothesis 2, we shall show how our algorithm maximises the number of target class embeddings.

**Statement 2:** Let the mean number of embeddings added within a particular interval of radius is given by $\lambda_1$ and $\lambda_2$ for target and non-target class distributions, respectively. As the non-target class contain all the $\mathbb{C} - 1$ classes, thus, $\lambda_2 >> \lambda_1$.

We validate our statement using paired t-test keeping the threshold for $p$-value at 0.01. We randomly select 10 clusters and calculate $\lambda_1$ and $\lambda_2$ values for each. The paired t-test is conducted on these set of values and hence, formulate the null hypothesis as:

$$H_0 : \mu_d = 0$$
$$H_1 : \mu_d \neq 0 \tag{5}$$

The $p$-value generated for this experiment is 0.00008. Therefore, we reject the null hypothesis and conclude that the difference between $\lambda_1$ and $\lambda_2$ is very high.

**Assumption:** If $x_{\lambda_1}$ and $x_{\lambda_2}$ be the points at which modes occur for both the distributions, respectively, then $x_{\lambda_1} < x_{\lambda_2}$.

Considering these statements and above hypotheses, we prove that our proposed cluster growing technique maximises the inclusion of target embeddings while minimising the non-target ones.

**Hypothesis 3:** The point of intersection of the Poisson distribution of the target and non-target embeddings is greater than the mode of the target Poisson distribution.

Let $f(x_1) = \frac{e^{-\lambda_1}\lambda_1^{x_1}}{x_1!}$ and $g(x_2) = \frac{e^{-\lambda_2}\lambda_2^{x_2}}{x_2!}$ be the distributions for target and non-target class embeddings, respectively. If $x_0$ is the point of intersection, then,

$$f(x_0) = g(x_0)$$

$$\ln f(x_0) = \ln g(x_0)$$

$$\ln e^{-\lambda_1} + \ln \lambda_1^{x_0} - \ln x_0! = \ln e^{-\lambda_2} + \ln \lambda_2^{x_0} - \ln x_0!$$

$$\ln e^{-\lambda_1} + \ln \lambda_1^{x_0} = \ln e^{-\lambda_2} + \ln \lambda_2^{x_0}$$

$$-\lambda_1 + x_0 \ln \lambda_1 = -\lambda_2 + x_0 \ln \lambda_2$$

$$x_0 = \frac{\lambda_2 - \lambda_1}{\ln \lambda_2 - \ln \lambda_1}$$

As $\lambda_2 >> \lambda_1$ thus, the point of intersection, $x_0$ becomes $\frac{\lambda_2}{\ln \lambda_2 - \ln \lambda_1}$.

To find the mode of the distribution we consider the following ratio for $x > 0$:

$$\frac{f(x+1)}{f(x)} = \frac{\frac{e^{-\lambda_1} \lambda_1^{x+1}}{(x+1)!}}{\frac{e^{-\lambda_1} \lambda_1^{x}}{x!}}$$

$$\frac{f(x+1)}{f(x)} = \frac{\lambda_1}{x+1}$$

Therefore, we observe that

$$\begin{aligned} f(x+1) > f(x) \quad \text{for} \quad x < \lambda_1 - 1 \\ f(x+1) < f(x) \quad \text{for} \quad x > \lambda_1 - 1 \end{aligned} \tag{6}$$

The maximum increase in the target embeddings occur when $x < \lambda_1 - 1$, and the point of intersection, $x_0 = \frac{\lambda_2}{\ln \lambda_2 - \ln \lambda_1} > \lambda_1 - 1$. The mode of the distribution for the target class embedding occurs before it intersects with the non-target class distribution. As $x_{\lambda_1} < x_{\lambda_2}$ (from assumption 3), we can conclude that our cluster growing technique, maximises the number of same class embeddings that are getting included in the discriminative cluster while the addition of non-target embeddings are limited.

### 3.2.2 Structure of the Clusters

We visualise the clusters formed in the embedding space using t-SNE plots as shown in Fig. 4. They form

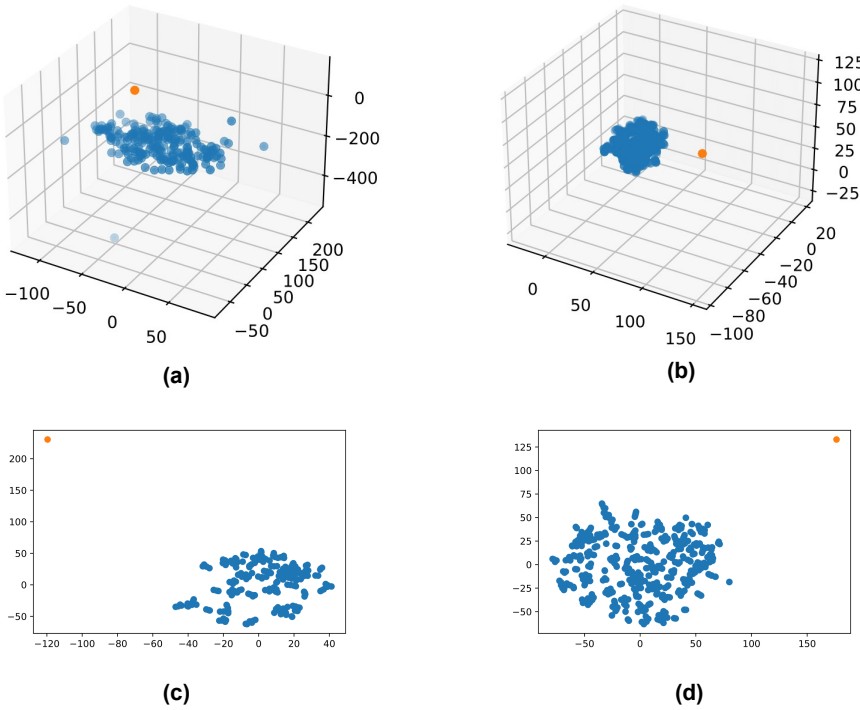

Figure 4: 3D and 2Ds plot of "airplane" and "automobile" class of *CIFAR10* using Canberra measure on *ViT* embeddings. The orange dot denote the centroid and blue dots represent the embeddings. (a) 3D plot of "airplane" class. (b) 3D plot of "automobile" class. (c) 2D plot of "airplane" class corresponding to (a). (d) 2D plot of "automobile" class corresponding to (b).

sectors with the cluster centroids $G$ as the centre. We mathematically denote each of these clusters based on the structure formed in Fig. 5. Empirically, we have observed that the space around the centroid is empty, and only after covering a minimum radius of $r_{min}$, we encounter the first embedding for that cluster. The

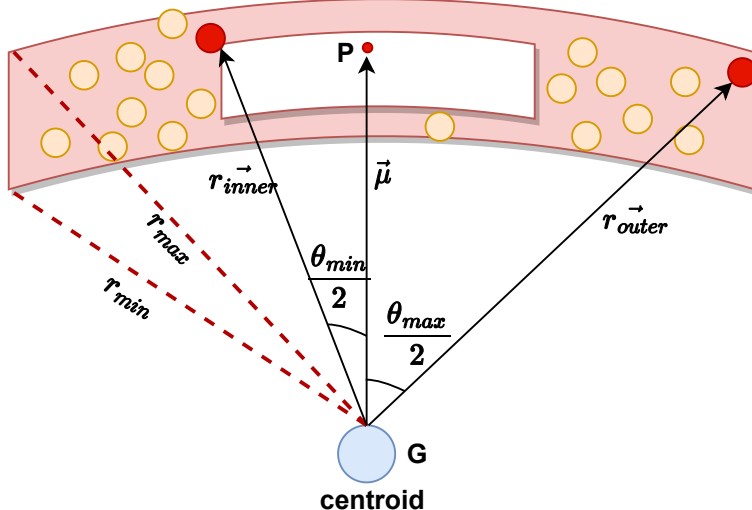

Figure 5: Structure of a cluster with mathematical notations.

number of embeddings follows the Poisson distribution and diminishes when the radius reaches the boundary of the cluster, $r_{max}$. Thus, all the embeddings of a cluster are concentrated between $r_{min}$ and $r_{max}$. We compute the mean vector $\vec{\mu_k}$ from the centroid $G_k$ for the cluster $k$ as:

$$\vec{\mu_k} = \frac{\sum_{i=1}^{n} e_{ik} - G_k}{n} \tag{7}$$

where, $e_{ik}$ represents the $i^{th}$ embedding of the $k^{th}$ cluster. Therefore, there exists a point $P_k$ within the cluster $k$ which can be denoted as $P_k = G_k + \vec{\mu_k}$. We designate this point as the *manifold mean point*, MMP. After calculating the mean vectors, we now compute the angle formed by the sector with respect to $\vec{\mu_k}$. We locate all the embeddings present at the boundary of the cluster with the soft threshold of 1. Thus, we form a set of embeddings present between $r_{max} - 1$ and $r_{max}$.

$$S_k = \{e_{ik} : r_{max} - 1 \leq e_{ik} - G_k \leq r_{max}, k \in \mathbb{C}\} \tag{8}$$

where, $e_{ik}$ and $G_k$ are the $i^{th}$ embedding and cluster centroid of the $k^{th}$ cluster, respectively.

We now compute the distance between each of the embeddings present in set $S_k$ and MMP $P_k$. The embedding with maximum distance from $P_k$ is considered as the point on the rim of the sector of $k^{th}$ cluster depicted as $e_{rim_k}$.

$$e_{rim_k} = argmax_j |e_{jk} - P_k|, e_{jk} \in S_k \tag{9}$$

Let $\vec{r_k} = e_{rim_k} - G_k$. Therefore, the angle $\frac{\theta_{max_k}}{2}$ formed between $\vec{\mu_k}$ and $\vec{r_k}$ is given by:

$$\frac{\theta_{max_k}}{2} = \arccos \frac{\vec{\mu_k} \cdot \vec{r_k}}{|\vec{\mu_k}||\vec{r_k}|} \tag{10}$$

Similarly, we compute $\frac{\theta_{min}}{2}$ and $\frac{\theta_{max}}{2}$ by considering the point at minimum and maximum distance from $P_k$, respectively. During visualization we have observed that the data points in a cluster lie between $\frac{\theta_{min}}{2}$ and $\frac{\theta_{max}}{2}$ which is calculated with respect to the centroid $G_k$ and $P_k$ for cluster $k$. Thus, it gives us an insight to the structure of a cluster in latent space. Both $G_k$ and $P_k$ are $n$-dimensional, and along with the minimum radius $r_{min}$, maximum radius $r_{max}$ and cluster angles, $\frac{\theta_{max}}{2}$ and $\frac{\theta_{min}}{2}$, form a $2n + 4$ parametric representation for each of the clusters. Table 2 shows the average minimum $\frac{\theta_{min}}{2}$ and maximum $\frac{\theta_{max}}{2}$ cluster angles (in radians) obtained for each dataset using various models and distance measures.

Table 2: Average minimum ($\frac{\theta_{min}}{2}$) and maximum ($\frac{\theta_{max}}{2}$) cluster angles (in radians) obtained for each dataset using various models and distance measures.

| Model | Distance function | CIFAR10 | | CIFAR100(20) | | CIFAR100 | | ImageNet | |
|---|---|---|---|---|---|---|---|---|---|
| | | min | max | min | max | min | max | min | max |
| ViT | Euclidean | 1.15±0.16 | 1.89±0.18 | 0±0.0 | 1.23±0.55 | 0±0.0 | 1.39±0.38 | 0±0.0 | 1.44±0.12 |
| | Manhattan | 1.36±0.08 | 1.76±0.11 | 0.94±0.19 | 1.48±0.21 | 0.58±0.23 | 1.68±0.25 | 0.77±0.08 | 1.31±0.10 |
| | Canberra | 1.31±0.10 | 1.71±0.13 | 0.62±0.33 | 1.65±0.32 | 0.43±0.13 | 1.84±0.32 | 0±0.0 | 1.61±0.10 |
| EfficientNetV2 | Euclidean | 0.91±0.07 | 2.33±0.11 | 0.78±0.14 | 2.45±0.16 | 0.97±0.10 | 2.10±0.15 | 0±0.0 | 2.20±0.23 |
| | Manhattan | 0.91±0.12 | 2.08±0.16 | 0.89±0.24 | 2.39±0.35 | 1.03±0.10 | 1.71±0.12 | 0.41±0.10 | 1.87±0.14 |
| | Canberra | 1.11±0.10 | 1.96±0.20 | 0.76±0.27 | 2.44±0.49 | 1.11±0.09 | 1.99±0.14 | 0±0.0 | 2.09±0.25 |
| ConvNeXt | Euclidean | 0.61±0.17 | 2.50±0.15 | 0.49±0.09 | 1.73±0.30 | 0.54±0.31 | 2.86±0.37 | 0±0.0 | 2.48±0.27 |
| | Manhattan | 0.71±0.28 | 2.07±0.22 | 0.61±0.18 | 1.57±0.27 | 0.53±0.31 | 2.37±0.35 | 0±0.0 | 2.36±0.32 |
| | Canberra | 0.45±0.43 | 2.08±0.26 | 0.77±0.38 | 2.26±0.35 | 0.46±0.37 | 1.97±0.33 | 0±0.0 | 2.39±0.35 |

### 3.2.3 Distance measures

The initial measure used to find the distance between embedding points for growing the cluster is Euclidean distance. It computes the shortest distance between any two points in the Euclidean space and is given by:

$$D(x, y) = \sqrt{\sum_{i=1}^{n} |x_i - y_i|^2} \tag{11}$$

We evaluate our method on two other distance measures, Manhattan and Canberra, to find the optimal metric that can define these clusters distinctly.

**Manhattan distance:** Manhattan distance is used to measure the distance between two real values vectors in high-dimensional data. It is given by:

$$D(x, y) = \sum_{i=1}^{n} |x_i - y_i| \tag{12}$$

**Canberra distance:** Canberra distance computes the distance between pair of vectors as a sum of series fraction differences between the coordinates of these objects. It is given by:

$$D(x, y) = \sum_{i=1}^{n} \frac{|x_i - y_i|}{|x_i| + |y_i|} \tag{13}$$

### 3.3 Constructing Class Relations tree

We generate the final clusters using Eqn. 3 and 4. Each of these clusters have one predominant class and multiple other classes. We compute the centroid of the final clusters as

$$G_k^f = \frac{\sum_{i=1}^{|X_k|} x_{ik}}{|X_k|} \tag{14}$$

where, $X_k$ is the set of all the embeddings present in cluster $k$. Using the $G_k^f$ for each of the clusters, we apply the unweighted pair group method with arithmetic mean (UPGMA) (Dawyndt et al., 2006) to build the relation trees. UPGMA is an agglomerative hierarchical clustering method that is commonly used to build phylogenetic trees. The algorithm constructs dendrograms that denote the structure present while computing the pair-wise similarity among the class embeddings. At every step, the two most similar clusters are grouped together to form an aggregate cluster.

The distance between any two clusters will be the average distance between all the pairs of objects present in those clusters. Let $x$ and $y$ be two objects present in cluster $A$ and $B$, respectively. Then the distance

between $A$ and $B$, is given by:

$$d(A, B) = \frac{1}{|A||B|} \sum_{x \in A} \sum_{y \in B} d(x, y) \tag{15}$$

where, $d(x, y)$ is the distance between the pair of objects, $x$ and $y$. If we introduce a new cluster, $M$, the distance between the aggregate cluster $A \cup B$ and $M$ will be computed as:

$$d(A \cup B, M) = \frac{|A|d(A, M) + |B|d(B, M)}{|A| + |B|} \tag{16}$$

We have used three different distance measures, Euclidean, Manhattan and Canberra, to construct the dendrograms using UPGMA algorithm.

## 4 Experimentation

The experiments have been divided into two parts: 1) studying the performance of the cluster growing technique. In this subsection, we check the quality of the clusters formed using different learning models; 2) examine the hierarchy present between the clusters. This subsection carefully studies and analyses the relationship among the classes, the overlaps and compares the relation trees formed with the existing *WordNet* hierarchy.

We perform the entire experiment on three different family of models, *ViT*, *EfficientNetV2* and *ConvNeXt*, to compare the types of embeddings generated by them. We summarise the steps of the experiments as follows:

- The performance of our proposed method is evaluated on *CIFAR10*, 20 coarse classes of *CIFAR100*, 100 fine classes *CIFAR100* (Krizhevsky et al., 2009) and *ImageNet* (Deng et al., 2009) data sets.

- The generated embeddings are grouped into distinct classes using our cluster growing technique. We denote them as class associated embedding clusters (CAEC).

- We analyse the quality of the clusters formed, and study the interaction between various classes using three different distance metrics

- The relationship among the classes are depicted in a tree structure, named *relation tree*, using the UPGMA algorithm (Dawyndt et al., 2006).

- The relation trees are then evaluated using phylogenetic tree comparison methods.

### 4.1 Data Set

The *CIFAR10* and *CIFAR100* (Krizhevsky et al., 2009) datasets are widely used for benchmarking algorithms in the field of computer vision. *CIFAR10* consist of $32 \times 32$ images denoting 10 classes. Each of the classes comprise of 5000 images for training and 1000 images for testing. *CIFAR100* has 100 classes with 600 samples for each class. Thus, a total of $50,000$ images are used in a training set. *CIFAR100* coarse data set form groups of 5 finer classes to form 20 resultant superclasses. We use these superclasses for our study and refer to it as *CIFAR100(20)*. We further use all the 100 classes in our experiments. The pre-trained model is fine-tuned on the training data for 50 epochs using stochastic gradient descent optimizer, keeping the learning rate fixed at 0.001. The test set containing $10,000$ images are then used to generate the embeddings. *ImageNet* (Deng et al., 2009) is the benchmark data set for image classification. It has 1.28 million images for training and $100,000$ for testing. As the models are already pre-trained on *ImageNet* data, we directly use the embeddings generated from the models.

### 4.2 Quality Analysis of Clusters

The cluster growing technique is applied using different distance measures. Each cluster formed encloses distinct representations of a particular class. Fig 6 shows a t-SNE plot of the clusters formed using Canberra distance applied on embeddings generated from *ViT* model.

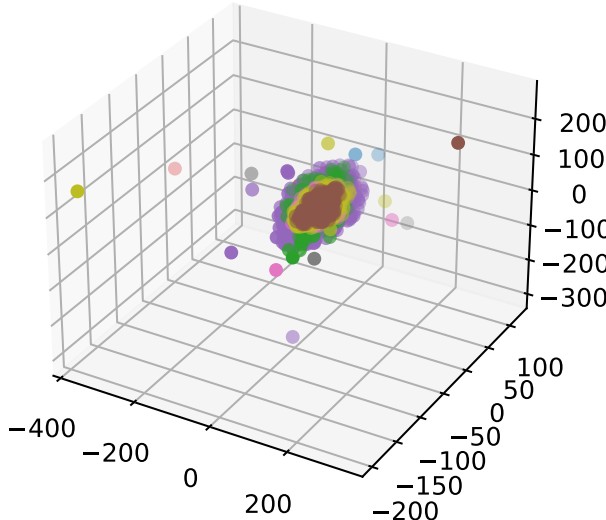

Figure 6: Cluster plot of *CIFAR10* using Canberra measure on *ViT* embeddings. airplane-blue, automobile-orange,

Table 3: Maximum radius obtained for all the classes of *CIFAR10* for various distance measures on embeddings from *ViT*, *EfficientNetV2* and *ConvNeXt*.

| Model | Distance function | airplane | automobile | bird | cat | deer | dog | frog | horse | ship | truck |
|---|---|---|---|---|---|---|---|---|---|---|---|
| | Euclidean | 203.28 | 209.32 | 191.96 | 186.97 | 194.40 | 190.06 | 203.37 | 195.40 | 199.25 | 203.56 |
| *ViT* | Manhattan | 54916 | 58842.66 | 53008.24 | 53027.37 | 53914.27 | 53045.04 | 53854.55 | 56783.96 | 51108.96 | 55887.05 |
| | Canberra | 81224 | 81441.78 | 79873.75 | 87293.50 | 82904.66 | 88326.50 | 78487.53 | 83287.62 | 78676.77 | 83474.0 |
| | Euclidean | 10.51 | 11.03 | 11.06 | 12.58 | 10.31 | 10.91 | 10.48 | 10.24 | 10.12 | 10.13 |
| *EfficientNetV2* | Manhattan | 277 | 240.96 | 261.42 | 295.78 | 262.08 | 272.39 | 241.22 | 243.21 | 240.65 | 229.34 |
| | Canberra | 550 | 537.72 | 574.53 | 644.60 | 560.61 | 602.65 | 534.70 | 511.98 | 496.56 | 514.62 |
| | Euclidean | 9.17 | 9.20 | 8.75 | 8.75 | 8.63 | 7.55 | 10.25 | 9.31 | 8.92 | 7.78 |
| *ConvNeXt* | Manhattan | 198 | 227.22 | 170.62 | 98.78 | 164.82 | 155.95 | 198.99 | 195.72 | 174.38 | 158.58 |
| | Canberra | 341 | 280.82 | 378.57 | 402.67 | 394.16 | 349.72 | 302.36 | 314.02 | 267.20 | 264.26 |

We observe annular structure for some clusters in the projected space using Fig. 6. In these cases, $\theta_{min}$ is observed to be greater than 45° or roughly 0.8 radian. Although the annular structure may not be prominent for individual clusters in the projected space, we assume clusters with $\theta_{min} \geq 0.8$ to follow this structure in high-dimensional latent space.

Some clusters may also contain few embeddings of different classes which are closely related or missclassified while training. We compute the radius of each cluster using our proposed method, and determine the coverage as fraction of target class embeddings to the total number of embeddings present in a cluster. Table 3 and 4 show the radius and coverage obtained using various distance measures on embeddings generated from different models on *CIFAR10*, respectively.

We note that the average coverage for classes like "cat" and "dog" is low for all the methods and high for "ship" and "truck." The average coverage for all other classes are comparable. However, the overall result observed in Table 4 using Canberra distance is better compared to the other two metrics. We obtain best results for embeddings generated using *EfficientNetV2*.

### 4.2.1 Cluster Purity

Each cluster is assigned a label based on the maximum occurring class embeddings. Purity of the cluster is estimated as the fraction of the number of matching class and cluster labels among the total number of embeddings present in all the clusters. The purity of a cluster with centre at $m_k$ is computed with respect to class $k$. Assuming $C$ to be the number of clusters formed and $N$ as the total number of embeddings present

Table 4: Coverage obtained for all the classes of *CIFAR10* for various distance measures on embeddings from *ViT*, *EfficientNetV2* and *ConvNeXt*.

| Model | Distance function | airplane | automobile | bird | cat | deer | dog | frog | horse | ship | truck |
|---|---|---|---|---|---|---|---|---|---|---|---|
| | Euclidean | 0.73 | 0.76 | 0.72 | 0.73 | 0.73 | 0.72 | 0.89 | 0.72 | 0.89 | 0.85 |
| *ViT* | Manhattan | 0.82 | 0.86 | 0.75 | 0.84 | 0.84 | 0.85 | 0.99 | 0.73 | 0.99 | 0.95 |
| | Canberra | 0.93 | **0.99** | **1.0** | **0.93** | 0.92 | 0.79 | **1.0** | 0.98 | **1.0** | **1.0** |
| | Euclidean | 0.98 | 0.98 | 0.97 | 0.81 | **0.98** | 0.92 | 0.98 | 0.99 | 0.99 | 0.99 |
| *EfficientNetV2* | Manhattan | 0.97 | 0.98 | 0.98 | 0.82 | 0.97 | 0.91 | 0.98 | 0.99 | 0.99 | 0.99 |
| | Canberra | **0.99** | 0.98 | 0.99 | 0.90 | 0.97 | 0.93 | 0.98 | 0.99 | 0.99 | 0.99 |
| | Euclidean | 0.90 | 0.98 | 0.79 | 0.52 | 0.85 | 0.75 | 0.90 | 0.96 | 0.98 | 0.97 |
| *ConvNeXt* | Manhattan | 0.88 | 0.96 | 0.81 | 0.72 | 0.91 | 0.76 | 0.92 | 0.97 | 0.99 | 0.98 |
| | Canberra | 0.97 | **0.99** | 0.93 | 0.74 | 0.92 | 0.86 | 0.98 | 0.99 | **1.0** | 0.98 |

in all the clusters, purity is given by:

$$purity = \frac{1}{N} \sum_{j}^{C} max_j |c_j \cap t_j| \tag{17}$$

where, $c_i$ is the cluster representing $i^{th}$ class and $t_j$ is the class embedding having maximum count for $c_i$. Table 5 compares the purity of the clusters obtained for embeddings generated from three family of models using different distance measures. We examine the average cluster purity obtained for each dataset using the three models in Table 6.

Table 5: Comparison of purity of clusters obtained for embeddings generated using *ViT*, *EfficientNetV2* and *ConvNeXt* using different distance metrics.

| Model | Distance function | CIFAR10 | CIFAR100(20) | CIFAR100 | ImageNet |
|---|---|---|---|---|---|
| | Euclidean | 0.80 | 0.35 | 0.39 | 0.04 |
| *ViT* | Manhattan | 0.86 | 0.66 | 0.57 | 0.07 |
| | Canberra | 0.95 | 0.83 | 0.71 | 0.11 |
| | Euclidean | 0.96 | 0.90 | **0.96** | **0.92** |
| *EfficientNetV2* | Manhattan | 0.97 | **0.94** | **0.96** | **0.92** |
| | Canberra | **0.98** | 0.86 | 0.91 | 0.88 |
| | Euclidean | 0.81 | 0.60 | 0.86 | 0.43 |
| *ConvNeXt* | Manhattan | 0.91 | 0.66 | 0.88 | 0.68 |
| | Canberra | 0.93 | 0.89 | 0.95 | 0.89 |

Table 6: Average cluster purity obtained for all the datasets using *ViT*, *EfficientNetV2* and *ConvNeXt* models.

| Model | CIFAR10 | CIFAR100(20) | CIFAR100 | ImageNet |
|---|---|---|---|---|
| *ViT* | 0.87±0.06 | 0.61±0.19 | 0.56±0.13 | 0.07±0.03 |
| *EfficientNetV2* | 0.97±0.01 | 0.90±0.03 | 0.94±0.02 | 0.91±0.02 |
| *ConvNeXt* | 0.88±0.05 | 0.72±0.14 | 0.90±0.04 | 0.67±0.19 |

### 4.2.2 Rand Index

Rand Index (RI) is a commonly used measure to find similarity among clustering methods by comparing the real labels with the cluster labels to evaluate the performance of an algorithm. It groups unordered data

points into pairs and matches the occurrences of each pair in the true and predicted clusters. For example, if we consider $x$ to be the number of pairs whose elements belong to the same cluster for both true and predicted labels, and $y$ to be the number of pairs whose elements do not belong to the same cluster for both true and predicted labels, RI is given by:

$$RI = \frac{x + y}{{}^{n}C_2} \tag{18}$$

where, ${}^{n}C_2$ form all pairs of unordered elements. Table 7 compares the RI of the clusters obtained for embeddings generated from three family of models using different distance measures.

Table 7: Comparison of Rand Index (RI) of clusters obtained for embeddings generated using *ViT*, *EfficientNetV2* and *ConvNeXt* using different distance metrics.

| Model | Distance function | CIFAR10 | CIFAR100(20) | CIFAR100 | ImageNet |
|---|---|---|---|---|---|
| | Euclidean | 0.92 | 0.62 | 0.95 | 0.97 |
| *ViT* | Manhattan | 0.94 | 0.85 | 0.95 | 0.96 |
| | Canberra | 0.98 | 0.95 | 0.97 | 0.97 |
| | Euclidean | **0.99** | 0.98 | **1.0** | **1.0** |
| *EfficientNetV2* | Manhattan | **0.99** | **0.99** | **1.0** | **1.0** |
| | Canberra | **0.99** | 0.97 | 0.99 | **1.0** |
| | Euclidean | 0.92 | 0.90 | 0.99 | 0.99 |
| *ConvNeXt* | Manhattan | 0.97 | 0.91 | 0.99 | 0.99 |
| | Canberra | 0.97 | 0.98 | **1.0** | **1.0** |

### 4.2.3 Normalised Mutual Information

Normalised Mutual Information (NMI) is a measure commonly used to assess network partitioning, and compare the partitions formed using community finding algorithms. It scales the value between 0 to 1, where NMI of 1 denote perfect correlation, while a value of 0 means no mutual information. Let the cluster and class labels be given by $C$ and $K$, respectively then NMI can be measured as:

$$NMI = \frac{2 \times I(K;C)}{H(K) \times H(C)} \tag{19}$$

where, $I(K;C)$ is the mutual information between $K$ and $C$, and $H(K)$ and $H(C)$ represent entropy of $K$ and $C$, respectively. Table 8 compares the NMI of the clusters obtained for embeddings generated from three family of models using different distance measures.

Table 8: Comparison of Normalised Mutual Information (NMI) of clusters obtained for embeddings generated using *ViT*, *EfficientNetV2* and *ConvNeXt* using different distance metrics.

| Model | Distance function | CIFAR10 | CIFAR100(20) | CIFAR100 | ImageNet |
|---|---|---|---|---|---|
| | Euclidean | 0.67 | 0.31 | 0.58 | 0.13 |
| *ViT* | Manhattan | 0.76 | 0.52 | 0.67 | 0.24 |
| | Canberra | 0.91 | 0.75 | 0.79 | 0.33 |
| | Euclidean | 0.93 | 0.84 | **0.97** | **0.97** |
| *EfficientNetV2* | Manhattan | 0.93 | **0.90** | **0.97** | **0.97** |
| | Canberra | **0.95** | 0.80 | 0.93 | 0.96 |
| | Euclidean | 0.71 | 0.47 | 0.89 | 0.60 |
| *ConvNeXt* | Manhattan | 0.82 | 0.55 | 0.92 | 0.81 |
| | Canberra | 0.88 | 0.84 | 0.96 | 0.95 |

From Table 5, 7 and 8, we observe that the quality of clusters formed using Manhattan distance on embeddings generated from *EfficientNetV2* is better based on the evaluation metrics that we have used. The results for *ViT* and *ConvNeXt* are comparable if we consider only the purity and RI measures when it comes to *CIFAR10* dataset. However, NMI of *ConvNeXt* surpasses *ViT* significantly. As the number of classes increase, the cluster purity of *ViT* decreases. We observe that for high dimensional data, *ViT* fail to capture feature similarity among classes. The overall cluster quality observed for all the embeddings are better when Manhattan or Canberra distances are used instead of Euclidean.

### 4.3 Paired t-test

We conduct paired t-test to statistically verify our analysis based on the results of coverage that we have observed using our cluster growing technique in Table 4. The $p$-values for every pair of distances are computed for each model embeddings. We define $\mu_d$ as the difference between the mean coverage between any two distance measure, and hence formulate the null hypothesis as:

$$H_0 : \mu_d = 0$$
$$H_1 : \mu_d \neq 0 \tag{20}$$

We fix the threshold for $p$-value at 0.01. Thus, two measures are giving significantly different results if $p$-value$\leq 0.01$. Otherwise, the results are comparable and we do not reject the null hypothesis. Table 9 show the results for paired t-test.

Table 9: $p$-values observed for the paired t-test.

| Model | Distance pairs | $p$-value | | | |
|---|---|---|---|---|---|
| | | *CIFAR10* | *CIFAR100(20)* | *CIFAR100* | *ImageNet* |
| *ViT* | Euclidean, Manhattan | **0.000041** | 0.362701 | 0.010575 | **0.000349** |
| | Euclidean, Canberra | **0.000016** | 0.439860 | 0.811339 | **7.30e-58** |
| | Manhattan, Canberra | 0.017404 | 0.969604 | 0.040832 | **1.28e-80** |
| *EfficienNetV2* | Euclidean, Manhattan | 0.678309 | **0.000068** | 0.152181 | 0.034765 |
| | Euclidean, Canberra | 0.217194 | 0.014037 | **0.000004** | **5.01e-23** |
| | Manhattan, Canberra | 0.134050 | **0.001735** | **0.000002** | **2.85e-23** |
| *ConvNeXt* | Euclidean, Manhattan | 0.171016 | 0.109041 | 0.134542 | **1.50e-66** |
| | Euclidean, Canberra | **0.005719** | **0.000076** | **0.000006** | **2.82e-120** |
| | Manhattan, Canberra | **0.008351** | **0.004372** | **0.000284** | **1.95e-30** |

From Table 9, we observe that for *ViT* model, Manhattan and Canberra measures are comparable when *CIFAR10* dataset is used. However, for *CIFAR100* with 20 coarse classes and 100 fine classes, no significant changes are observed. On the other hand, clustering using Euclidean and Manhattan distances show comparable cluster coverage on all the datasets except *ImageNet* for *ConvNeXt* model. In general, when *ImageNet* dataset is used, the mean difference between the coverage using all distance measures are incomparable. In case of *EfficientNetV2*, we observe that as the number of classes increase, Euclidean measure show significant variation in coverage when compared to Manhattan and Canberra measures.

### 4.4 Interaction among Classes

The classes present in the datasets are highly correlated and can be grouped under an aggregated or parent class. The semantic similarity among the classes may vary based on their lowest common ancestor. Due to the inherent relation that exist among them, trained models tend to misclassify similar target labels or form representations which are ambiguous to both the classes. These embeddings affect the performance of the model on unknown data as they are not well defined and discriminative. We conduct a detailed study of these correlations, and build a tree-like structure depicting the class relations using the *CIFAR10*, *CIFAR100(20)*, *CIFAR100* and *ImageNet* dataset.

The proposed clustering algorithm may not provide partitioning of the space and manifolds of clusters may overlap. This implies that some of the embeddings may have ambiguous assignment of multiple classes. This kind of interaction is called "intrusion". On the other hand, there may be an embedding of an instance of a class "B" belonging to the cluster of a class "A". This is a case of misclassification and we call this interaction "infiltration". Next we observe the set of intrusive and infiltrated classes given a cluster of a specific class.

We have divided our study into two parts: 1) Section 4.4.1: In this Section, we study the presence of infitrated classes. We group non-target embeddings which are part of the cluster formed by a target label using our cluster growing technique. These embeddings are not present in their respective original clusters and hence, are misclassified. 2) Section 4.4.2: We study the presence of intrusive classes whose embeddings are ambiguous and part of multiple clusters denoting distinct classes. They form the overlapping region among two clusters. Such interaction is mostly seen in highly correlated classes. The representations learnt by the model cannot identify a particular class with high confidence as they are considered discrete by more than one class. Both the studies have been performed on all the classes. We show the results observed on all or randomly selected 10 classes (whichever is less) for all the datasets.

### 4.4.1 Presence of infiltrated classes

Each cluster contains embeddings of other classes which are either highly similar to the class in concern or have infiltrated the cluster due to misclassification. Typically, these representations are not present in their respective distinct clusters as they are not discriminative. We identify these classes, and group them for each cluster. Our algorithm tries to minimise the inclusion of other class embeddings when we increase the radius by a unit distance. Table 10, 11, 12 and 13 show the top 3 classes that are present in each distinct cluster based on the number of embeddings ($>= 2$) using Euclidean, Manhattan and Canberra distances for *CIFAR10*, *CIFAR100(20) CIFAR100* and *ImageNet*, respectively.

From Table 10, 11, 12 and 13, we observe that maximum infiltration happens using embeddings generated from *ViT* model. Considering the average inclusion of other classes through all the datasets, we notice least infiltration when Canberra distance is used for clustering. If we examine the results of each of the models, we note that *EfficientNetV2* has shown least interaction when Manhattan or Canberra metric is used. Moreover, the classes included for all the distance measures are highly similar to the cluster.

However, for *EfficientNetV2* and *ConvNeXt*, unfamiliar classes are grouped, with *ConvNeXt* showing maximum discrepancy when Euclidean measure is used. For example, in Table 10 if we consider the cluster denoting "automobile" class, the group of class present for *ViT* are "truck", "ship" and "airplane" for Euclidean. On the other hand, *EfficienNetV2* and *ConvNeXt* have included "cat" and "dog," respectively. All the clusters of *ConvNeXt* model when computed using Euclidean measure have incorporated the embeddings of "cat." The results show vast improvement when Manhattan and Canberra distances are used. All the three models group semantically related classes for each of the clusters.

The interaction among the classes start decreasing with the increase in the total number of classes. We observe that the clusters formed are more distinct and have very less embeddings from non-target class. *ImageNet*, although with 1000 classes, show less infiltration when compared to *CIFAR10* with only 10 classes. One of the main reasons behind this trend is the number of samples with which the model is trained for each of the classes. Thus, the model is able to learn more discriminative representations for each of the classes. Moreover, these large models are typically devised to work well on *ImageNet*, and are mostly fine-tuned on small-scale datasets.

### 4.4.2 Presence of intrusive classes

Overlapping regions are observed when embeddings are shared by multiple distinct clusters. Thus, a representation of class "airplane" may be present in both "airplane" and "ship" clusters. In Section 4.4.1, we identify those classes which are present in a different embedding cluster. They are not part of their original cluster. However, in this case an overlap among the clusters are observed. Hence, a particular embedding is recognised as a discrete representative by more than one classes. Table 14 represents the embeddings of those classes which occupy more than one clusters from *ViT*, *EfficientNetV2* and *ConvNeXt* for *CIFAR10* dataset.

Table 10: Top 3 most occurring infiltrated classes in a given cluster for all the models using Euclidean, Manhattan and Canberra distances on *CIFAR10* dataset. The number inside the brackets show the number of embeddings of that class present in the given cluster.
Left: Using Euclidean distance, Right: Using Manhattan distance, Bottom left: Using Canberra distance.

| Clusters | ViT | EfficientNetV2 | ConvNeXt |
|---|---|---|---|
| airplane | ship (67)
bird (34)
truck (22) | ship (2)
-
- | cat (14)
bird (10)
ship (3), frog (3) |
| automobile | truck (169)
ship (14)
airplane (4) | truck (4)
cat (2)
- | cat (4)
truck (3)
- |
| bird | deer (17)
airplane (6)
frog (5) | cat (7)
deer (3)
airplane (2) | cat (54)
deer (23)
frog (17) |
| cat | dog (14)
deer (3)
bird (2), frog (2) | dog (17)
frog (2)
deer (1), bird (1) | dog (61)
bird (24)
frog (17) |
| deer | bird (38)
cat (23)
dog (20) | cat (6)
bird (3)
horse (2) | cat (58)
bird (27)
frog (12) |
| dog | cat (47)
deer (8)
bird (7) | cat (63)
bird (3)
- | cat (221)
deer (11)
bird (10) |
| frog | bird (25), deer (25)
cat (18)
dog (6) | cat (5)
-
- | cat (4)
bird (17)
deer (8) |
| horse | deer (32)
dog (11)
cat (6) | cat (7)
dog (2)
- | cat (53)
dog (9)
deer (8) |
| ship | airplane (46)
truck (5)
automobile (2) | cat (2)
-
- | airplane (12)
cat (5)
dog (3) |
| truck | automobile (39)
airplane (26)
ship (25) | automobile (5)
-
- | cat (14)
airplane (7)
automobile (4), bird (4) |

| Clusters | ViT | EfficientNetV2 | ConvNeXt |
|---|---|---|---|
| airplane | ship (43)
bird (28)
truck (15) | ship (2)
-
- | bird (7)
frog (2), ship (2)
- |
| automobile | truck (103)
ship (3)
airplane (2) | truck (2)
-
- | airplane (2)
-
- |
| bird | airplane (11)
deer (4)
frog (2) | cat (6)
airplane (4), deer (4)
- | airplane (16)
deer (13), frog (13)
dog (5) |
| cat | dog (9)
deer (2)
- | dog (19)
deer (2), frog (2)
- | dog (57)
bird (17)
frog (11) |
| deer | bird (20)
cat (10)
frog (7) | cat (4)
horse (2)
- | bird (18)
frog (8)
dog (4), horse (4) |
| dog | cat (23)
bird (3)
deer (2), horse (2) | cat (35)
bird (3)
deer (2) | bird (9)
cat (6)
frog (5) |
| frog | -
-
- | cat (4)
-
- | bird (10)
deer (6)
airplane (4) |
| horse | deer (104)
dog (30)
cat (16) | dog (5)
cat (4)
- | dog (9)
airplane (6)
deer (4) |
| ship | -
-
- | airplane (2)
-
- | airplane (17)
dog (4)
automobile (3) |
| truck | automobile (13)
airplane (10)
ship (4) | automobile (4)
airplane (2)
- | automobile (11)
airplane (8)
automobile (4), bird (4) |

| Clusters | ViT | EfficientNetV2 | ConvNeXt |
|---|---|---|---|
| airplane | bird (14)
ship (2) | ship (2)
- | bird (2), deer (2)
- |
| automobile | truck (3) | truck (3) | truck (2) |
| bird | -
-
- | deer (5)
cat (4)
dog (2) | deer (17)
cat(5)
frog (2) |
| cat | dog (11)
frog (7)
deer (1) | dog (17)
deer (3), frog (3)
- | dog (41)
bird (7)
deer (5) |
| deer | bird (21)
horse (4)
frog (3) | -
-
- | bird (14)
cat (10)
dog (2) |
| dog | cat (49)
frog (21)
deer (15) | cat (25)
deer (2)
- | cat (88)
deer (5)
bird (3) |
| frog | -
-
- | deer (2)
-
- | cat (8)
deer (7)
bird (5) |
| horse | deer (4)
-
- | dog (2)
-
- | dog (8)
deer (7)
cat (5) |
| ship | - | - | airplane (4) |
| truck | - | automobile (7) | bird (2), cat (2) |

Table 14 show the number of cluster embeddings present in multiple distinct clusters. The third column represent the distinct clusters, while the fourth column group all the other classes where embeddings from the distinct cluster is present. For example, six embeddings of "airplane" cluster are also part of the "bird" cluster.

Table 11: Top 3 most occurring infiltrated classes in a given cluster for all the models using Euclidean, Manhattan and Canberra distances on *CIFAR100(20)* dataset. The number inside the brackets show the number of embeddings of that class present in the given cluster.

aq mammals: aquatic mammals, non-inst invtb: non-insect invertebrates, fr veg :fruit and vegetables, hed: household electrical device, containers: food containers, sm: small mammals, hsld furtr: household furniture, omni-herb: large omnivores and herbivores, mm: medium-sized mammals, mm outdr: large man-made outdoor things, nos: large natural outdoor scenes, lc: large carnivores.

Left: Using Euclidean distance, Right: Using Manhattan distance, Bottom left: Using Canberra distance.

| Clusters | ViT | EfficientNetV2 | ConvNeXt |
|---|---|---|---|
| fish | - 
 - 
 - | reptiles (7) 
 aq mammals (6) 
 non-inst invtb (2) | - 
 - 
 - |
| fr veg | - | containers (2) | - |
| hed | - 
 - 
 - | containers (17) 
 hsld furtr (9) 
 vehicles 2 (3) | containers (3) 
 - 
 - |
| hsld furtr | - 
 - 
 - | hed (11) 
 mm outdr (3) 
 vehicles 2 (2) | hed (2) 
 - 
 - |
| non-inst invtb | - 
 - 
 - | insects (19) 
 reptiles (17) 
 fr veg (4) | sm (8) 
 nos (5) 
 - |
| reptiles | non-inst invtb (15) 
 mm (14) 
 sm (13) | non-inst invtb (11) 
 fish (8) 
 omni-herb (6) | sm (4) 
 non-inst invtb (3) 
 - |
| sm | - 
 - 
 - | mm (28) 
 omni-herb (9) 
 aq mammals (6) | lc (4) 
 non-inst invtb (3) 
 - |
| tree | - 
 - | nos (6) 
 flowers (2) | - 
 - |
| vehicles 1 | - 
 - | vehicles 2 (18) 
 - | - 
 - |
| vehicles 2 | - 
 - 
 - | vehicles 1 (27) 
 mm outdr (6) 
 - | - 
 - 
 - |

| Clusters | ViT | EfficientNetV2 | ConvNeXt |
|---|---|---|---|
| fish | - 
 - | aq mammals (2) 
 sm (2) | - 
 - |
| fr veg | - | flowers (2) | - |
| hed | - 
 - | containers (11) 
 hsld furtr (6) | - 
 - |
| hsld furtr | - 
 - | - 
 - | nos (2) 
 - |
| non-inst invtb | - 
 - 
 - | insects (5) 
 reptiles (3) 
 - | - 
 - 
 - |
| reptiles | - 
 - 
 - | non-inst invtb (9) 
 fish (7) 
 aq mammals (5) | non-inst invtb (7) 
 mm (6) 
 - |
| sm | - 
 - 
 - | mm (3) 
 - 
 - | mm (5) 
 lc (4) 
 - |
| trees | nos (5) 
 mm outdr (4) 
 - | - 
 - 
 - | - 
 - 
 - |
| vehicles 1 | - 
 - 
 - | vehicles 2 (8) 
 - 
 - | - 
 - 
 - |
| vehicles 2 | - 
 - 
 - | vehicles 1 (17) 
 mm outdr (3) 
 - | - 
 - 
 - |

| Clusters | ViT | EfficientNetV2 | ConvNeXt |
|---|---|---|---|
| fish | aq mammals (18) 
 reptiles (10) 
 insects (9) | - 
 - 
 - | aq mammals (9) 
 - 
 - |
| fr veg | containers (8) 
 fish (3) 
 insects (2) | flowers (11) 
 containers (4) 
 - | - 
 - 
 - |
| hed | containers (14) 
 hsld furtr (6) 
 fish (3) | containers (11) 
 hsld furtr (7) 
 - | hsld furtr (13) 
 containers (11) 
 - |
| hsld furtr | containers (13) 
 hed (12) 
 fish (10) | - 
 - 
 - | hed (8) 
 - 
 - |
| non-inst invtb | fish (49) 
 container (2) 
 - | insects (5) 
 fr veg (3) 
 - | reptiles (5) 
 - 
 - |
| reptiles | fish (75) 
 mm (18) 
 insects (17) | - 
 - 
 - | aq mammals (15) 
 non-inst invtb (13) 
 fish (12) |
| sm | mm (53) 
 fish (12) 
 omni-herb (7) | people (10) 
 mm (7) 
 lc (6) | aq mammals (18) 
 mm (14) 
 fish (5) |
| tree | nos (14) 
 mm outdr (6) 
 insects (2) | - 
 - 
 - | - 
 - 
 - |
| vehicles 1 | vehicles 2 (10) 
 mm outdr (7) 
 nos (3) | vehicles 2 (21) 
 mm outdr (5) 
 hsld furtr (4) | vehicles 2 (10) 
 hsld furtr (7) 
 - |
| vehicles 2 | vehicles 1 (22) 
 fish (7) 
 insects (4) | vehicles 1 (53) 
 trees (5) 
 mm outdr (4) | vehicles 1 (33) 
 mm outdr (12) 
 - |

Table 12: Top 3 most occurring infiltrated classes in a given cluster for all the models using Euclidean, Manhattan and Canberra distances on *CIFAR100* dataset. The number inside the brackets show the number of embeddings of that class present in the given cluster.
Left: Using Euclidean distance, Right: Using Manhattan distance, Bottom left: Using Canberra distance.

| Clusters | ViT | EfficientNetV2 | ConvNeXt |
|---|---|---|---|
| beaver | otter (2) | porcupine (2) | - |
| clock | bowl (2), plate (2) | - | - |
| cloud | plain (3) | mountain (2), sea (2) | sea (4) |
| | rocket (2), sea (2) | - | - |
| dinosaur | crocodile (2) | - | woman (4) |
| | - | - | man (3) |
| | - | - | elephant (2) |
| dolphin | whale (3) | whale (4) | shark (3) |
| mouse | shrew (2) | - | - |
| mushroom | squirrel (2), beaver (2) | - | |
| plain | sea (2) | sea (2) | sea (2) |
| poppy | tulip (3) | - | - |
| ray | caterpillar (2) | - | flatfish (7) |
| | - | - | man (2), shark (2) |
| rose | tulip (6) | - | - |
| | poppy (4) | - | - |

| Clusters | ViT | EfficientNetV2 | ConvNeXt |
|---|---|---|---|
| beaver | - | porcupine (3) | porcupine (2) |
| beetle | cockroach (3) | - | cockroach (2) |
| clock | bowl (2), plate (2) | - | - |
| cloud | - | - | sea (6) |
| dolphin | - | whale (3) | - |
| poppy | - | - | tulip (4) |
| ray | - | - | flatfish (7) |
| | shark (5) | - | |
| | caterpillar (2), rabbit (2) | - | - |
| skyscraper | rocket (6) | - | - |
| | castle (3) | - | - |
| trout | crocodile (6) | - | - |
| | caterpillar (4) | - | - |

| Clusters | ViT | EfficientNetV2 | ConvNeXt |
|---|---|---|---|
| beaver | cockroach (2) | lion (2) | - |
| bed | chair (9) | - | couch (2) |
| | trout (4) | - | - |
| beetle | cockroach (18) | - | - |
| clock | telephone (8) | - | - |
| | plate (5) | - | - |
| | chair (3) | - | - |
| cloud | - | mountain (2) | sea (7) |
| dolphin | shark (2) | - | - |
| lamp | cup (19) | cup (6) | - |
| | bottle (7) | - | - |
| | cockroach (6) | - | - |
| mouse | hamster (5) | - | shrew (4) |
| | - | - | hamster (3) |
| otter | trout (8), skunk (8) | - | - |
| | cockroach (7), dinosaur (7) | - | - |
| | whale (6) | - | - |
| plain | - | - | sea (4) |
| ray | shark (10) | - | - |

We observe that maximum overlap occurs when embeddings from *ViT* model is used followed by *ConvNeXt* and *EfficientNetV2*. Although the overlap decreases using Canberra distance, it is still significantly more compared to the other models. The embeddings from *EfficientNetV2* produce well partitioned clusters. Similar trends are detected using Manhattan and Canberra distances for *ConvNeXt*. However, overlaps with "cat" cluster is present when Euclidean measures are used.

From Table 14, we observe that the models, *EfficientNetV2* and *ConvNeXt* show minimum overlapping clusters. Therefore, to further analyse the difference between these models, we select a few classes from *CIFAR100* and *ImageNet* datasets, and list all the overlapping embeddings for those classes using these two model embeddings in Table 15 and 16, respectively.

From Table 15 and 16, we observe that for the selected classes, the number of overlapping embeddings is less in case of *ConvNeXt* model. However, this may vary when another set of classes are chosen. In general, using Canberra metric has substantially reduced the overlapping regions for both the models.

## 4.5 Comparison of Relation trees

We generate the relation trees using two parameters 1) cluster centroids $G$ for each of the class representations, and 2) point through which the mean vector $\vec{\mu}$ from the centroid $G$ passes through the cluster denoted as the MPP, $P = G + \vec{\mu}$. The trees are generated by applying the UPGMA algorithm (Dawyndt et al., 2006).

Table 13: Top 3 most occurring infiltrated classes in a given cluster for all the models using Euclidean, Manhattan and Canberra distances on *ImageNet* dataset. The number inside the brackets show the number of embeddings of that class present in the given cluster.
husky: Siberian husky, A. terrier: Australian terrier, B. terrier: Bedlington terrier, setter: English setter, springer: English springer, t.t. sloth: three toed sloth, b.f. ferret: black footed ferret, M. hairless: Mexican hairless, cockatoo: sulphur crested cockatoo, camel: Arabian camel, cobra: Indian cobra, fbhelmet: football helmet, stocking: Christmas stocking, rb sandpiper : red backed sandpiper, b. swan: black swan, cellphone: cellular telephone, tie: Windsor tie.
Left: Using Euclidean distance, Right: Using Canberra distance, Bottom left: Using Manhattan distance.

| Clusters | ViT | EfficientNetV2 | ConvNeXt |
|---|---|---|---|
| husky | A. terrier (5) | - | setter (4) |
| | grey whale (4), Airedale (4) | - | A. terrier (2) |
| | springer (3), setter (3) | - | - |
| A. terrier | huskey (8) | - | - |
| | hartebeest (5), collie (5) | - | - |
| | guacamole (4), spider web (4) | - | - |
| t.t. sloth | skunk (3), coyote (3) | - | - |
| | B. terrier (2) | - | - |
| M.hairless | - | b.f. ferret (14) | b.f. ferret (33) |
| mousetrap | collie (7) | cockatoo (4) | - |
| | Pomerarian (6), camel (6), siamang (6) | - | - |
| | whiskey jug (5), pop bottle (5) | - | - |
| broccoli | spaghetti squash (5), running shoe (5) | - | king penguin (2) |
| | scoreboard (4), plate rack (4) | - | - |
| | shoe shop (3), tobacco shop (3) | - | - |
| safe | printer (8) | - | spider web (2) |
| | coucal (7), hummingbird (7), spoonbil (7) | - | mailbag (2) |
| | black stork (5), box turtle (5) | - | - |
| fbhelmet | coucal (11), hummingbird (11), spoonbil (11), | - | mailbag (2) |
| | bee eater (10), b. swan (10), rb sandpiper (10) | - | - |
| | goose (9), bittern (9) | - | - |
| sarong | tie (8) | - | overskirt (2) |
| | volleyball (7) | - | - |
| | overskirt (5) | - | - |
| pretzel | quilt (5), tape player (5) | - | mud turtle (17) |
| | mud turtle (4), bannister (4) | - | tench (11) |
| | bulletproof vest (3), agaric (3) | - | bannister (8) |

| Clusters | ViT | EfficientNetV2 | ConvNeXt |
|---|---|---|---|
| sundial | - | parking meter (16) | parking meter (2) |
| | - | toaster (12) | - |
| neck brace | junco (26) | - | - |
| | ocarina (25), flamingo (25) | - | - |
| wolf spider | megalith (28) | partridge (2) | - |
| | slug (26) | - | - |
| stone wall | tile roof (30) | - | - |
| | cellphone (22) | - | - |
| | harvester (19) | - | - |
| stocking | bulbul (24) | - | - |
| | Siberian husky (22) | - | - |
| | Australian terrier (19) | - | - |
| mitten | vine snake (45) | vine snake (2) | - |
| | cellphone (43) | - | - |
| | tile roof (39) | - | - |
| pretzel | prairie chicken (34) | - | - |
| | basketball (20) | - | - |
| | peacock (16), castle (16) | - | - |

| Clusters | ViT | EfficientNetV2 | ConvNeXt |
|---|---|---|---|
| t.t. sloth | skunk (2), coyote (2) | - | - |
| tench | mud turtle (7), bannister (7), agaric (7) | - | pretzel (5) |
| | loupe (6), quilt (6) | - | loupe (5) |
| sidewinder | sea anemone (6), drumstick (6), lotion (6) | lotion (2) | mailbag (2) |
| altar | cobra (8), spiny lobster (8) | Petri dish (2) | - |
| | green mamba (7), rule (7) | - | - |
| fbhelmet | drake (2), spider web (2) | - | - |
| drumstick | sea anemone (6), cucumber (6) | - | - |
| | radiator (4), combination lock (4) | - | - |
| sarong | tie (3) | - | - |
| | parachute (2) | - | - |

All the three distance metrics, Euclidean, Manhattan and Canberra, are used in UPGMA (Dawyndt et al., 2006) to generate the dendrograms. As a reference tree, we use a sub-tree derived from the *WordNet* (Miller, 1995) ontology. Each leaf node depicts a particular class present in the datasets. The hierarchy is built using semantic relationship among the classes present (Bertinetto et al., 2020). We compare the relation trees formed by *ViT*, *EfficientNetV2* and *ConvNeXt* with the *CLIP* (Radford et al., 2021) model. *CLIP* learn visual representations from natural language supervision using joint learning of image and text pairs. Fig. 7 show the dendrograms formed before and after the clustering algorithm is applied using Canberra distance for *CIFAR10* dataset.

### 4.5.1 Robinson-Foulds distance metric

Robinson-Foulds (Robinson & Foulds, 1981) is a widely used metric to find the distance between trees by comparing the number of splits that differ for a pair of tree. Each branch is removed and the number of partitions unique to the tree is calculated. The total number of such partitions between a pair of trees form the Robinson-Foulds distance. We calculate Robinson-Foulds (RF) distances for the generated trees with respect to the *WordNet* reference tree in Table 17 using centroid $G$ and manifold mean point $P$.

Table 14: List of intrusive classes for a cluster of a specific class using *CIFAR10* dataset. The number inside the brackets show the number of overlapping embeddings present.

| Model | Distance function | Class-specific cluster | Overlapping Embeddings |
|---|---|---|---|
| ViT | Euclidean | airplane | ship (45), truck (25), bird (6) |
| | | automobile | truck (39), airplane (3), ship (2) |
| | | bird | deer (29), frog (19), airplane (15) |
| | | cat | dog (28), deer (9), frog (7) |
| | | deer | horse (30), frog (25), bird (17) |
| | | dog | deer (15), cat (14), horse (9) |
| | | frog | deer (14), dog (6), bird (5) |
| | | horse | deer (15), airplane (4), dog (3) |
| | | ship | airplane (63), truck (18), automobile (7) |
| | | truck | automobile (156), airplane (20), ship (5) |
| | Manhattan | airplane | bird (11), dog (11), truck (11) |
| | | automobile | truck (11), airplane (1) |
| | | bird | deer (10), airplane (9), horse (4) |
| | | cat | dog (20), horse (8), airplane (3) |
| | | deer | horse (75), bird (4), airplane (2) |
| | | dog | horse (17), cat (7), airplane (3) |
| | | frog | deer (4), bird (1) |
| | | horse | deer (6), dog (2), airplane (1) |
| | | ship | airplane (25) |
| | | truck | automobile (71), airplane (12) |
| | Canberra | bird | deer (7), dog (6), airplane (2) |
| | | cat | dog (38) |
| | | deer | dog (13), horse (4), cat (1) |
| | | dog | cat (11) |
| | | frog | dog (10), deer (1) |
| | | horse | deer (2) |
| | | ship | airplane (25) |
| | | truck | automobile (2) |
| EfficientNetV2 | Euclidean | dog | cat (2) |
| | Manhattan | - | - |
| | Canberra | - | - |
| ConvNeXt | Euclidean | bird | cat (4) |
| | | cat | dog (26), deer (4), bird (2) |
| | | deer | cat (6) |
| | | dog | cat (54) |
| | Manhattan | dog | cat (1) |
| | Canberra | - | - |

The the three columns for each dataset in Table 17 denote the distance measure used to generate the trees. We observe best result on *CIFAR10* for *ViT* using Canberra distance for cluster formation, and Euclidean measure for tree generation. *EfficientNetV2* show better results on *CIFAR100* coarse and fine when Canberra and Manhattan distances are used, respectively. The results for *ImageNet* dataset remain almost indistinguishable from before and after cluster formation. The trend remains the same when trees are generated using centroid $G$ or manifold mean point $P$. The observed Robinson-Foulds distance using $G$ and $P$ are almost similar. However, the average results are slightly better when $P$ is considered for UPGMA.

Table 15: List of intrusive classes for a cluster of a specific class using *CIFAR100* dataset on *EfficientNetV2* and *ConvNeXt* model on the same set of clusters. The number inside the brackets show the number of overlapping embeddings present.

| Model | Distance function | Class-specific cluster | Overlapping Embeddings |
|---|---|---|---|
| EfficientNetV2 | Euclidean | apple | orange (5) |
| | | baby | boy (18), girl (4) |
| | | man | boy (15), woman (8) |
| | | bear | seal (4) |
| | | beaver | seal (6), otter (4) |
| | | bed | couch (16) |
| | | oak tree | maple tree (26), pine tree (18), willow tree (10) |
| | | tractor | lawn mower (13) |
| | Manhattan | apple | sweet pepper (4) |
| | | baby | boy (10), girl (3) |
| | | man | boy (22) |
| | | bear | seal (5), otter (1) |
| | | beaver | seal (4), otter (4) |
| | | bed | couch (7) |
| | | oak tree | pine tree (16), willow tree (13), maple tree (10) |
| | | tractor | lawn mower (4) |
| | Canberra | apple | sweet pepper (1) |
| | | baby | girl (7) |
| | | man | woman (13) |
| | | bed | couch (45) |
| | | oak tree | maple tree (49), willow tree (25) |
| | | tractor | lawn mower (13) |
| ConvNeXt | Euclidean | apple | pear (4) |
| | | baby | boy (1), caterpillar (1) |
| | | man | boy (28), bowl (5), girl (2) |
| | | bed | couch (6) |
| | Manhattan | apple | pear (4) |
| | | baby | boy (10), bowl (2), flatfish (2) |
| | | man | boy (22) |
| | | bed | couch (2) |
| | | oak tree | willow tree (1) |
| | Canberra | beaver | mouse (2) |
| | | oak tree | pine tree (1) |

### 4.5.2 Deformity Index

There exists unresolved relationship among species due to which it is difficult to acquire an accurate reference tree for any dataset. Thus, prevalent tree comparison techniques may show unexpected results under different circumstances. Deformity index (Mahapatra & Mukherjee, 2021) is a scoring system that measures the dissimilarity among different phylogenetic trees based on the list of clades given in a reference tree. The measure is dependent on the list of clades obtained either from the reference tree or hypotheses. Deformity index of the tree $T$ is given by:

$$D(T) = \frac{1}{|\Lambda(T_R)|} \sum_i Dc(\Lambda_i); \forall \Lambda_i \in \Lambda(T_R) \tag{21}$$

Table 16: List of intrusive classes for a cluster of a specific class using *ImageNet* dataset on *EfficientNetV2* and *ConvNeXt* model on the same set of clusters. The number inside the brackets show the number of overlapping embeddings present.

slipper: yellow lady's slipper, gn schnauzer: giant schnauzer, S. terrier: Sealyham terrier, B. griffon: Brabancon griffon, M. hairless: Mexican hairless, I. wolfhound: Irish wolfhound, bf. ferret: black-footed ferret, chicken: prairie chicken, h. monkey: howler monkey

| Model | Distance function | Class-specific cluster | Overlapping Embeddings |
|---|---|---|---|
| EfficientNetV2 | Euclidean | titi | schnauzer (24), meerkat (8) |
| | | bf. ferret | M. hairless (39), skunk (4) |
| | | Saluki | schipperke (289) |
| | | schipperke | Saluki (297) |
| | | S. terrier | B. griffon (287) |
| | | h.monkey | hippopotamus (48) |
| | | ambulance | beach wagon (50) |
| | | tractor | mobile home (283) |
| | | trombone | French horn (83) |
| | | cliff | slipper (46) |
| | | chicken | basketball (50) |
| | | dishwasher | artichoke (154) |
| | | artichoke | sea urchin (154) |
| | | earthstar | sunscreen (68) |
| | | bubble | espresso (66) |
| | Manhattan | titi | schnauzer (23), meerkat (11) |
| | | bf. ferret | M. hairless (17) |
| | | schipperke | Saluki (4), colobus (3) |
| | | S. terrier | I. wolfhound (280) |
| | | tractor | mobile home (280) |
| | | trombone | French horn (33) |
| | | chicken | basketball (55) |
| | | artichoke | sea urchin (5) |
| | | earthstar | sunscreen (25) |
| | | bubble | espresso (11), gown (4) |
| | Canberra | titi | schnauzer (16), meerkat (13) |
| | | bf. ferret | M. hairless (6) |
| | | S. terrier | wild boar (4), wool (3) |
| | | tractor | mobile home (3) |
| | | trombone | French horn (11) |
| | | chicken | basketball (13) |
| | | earthstar | sunscreen (28) |
| | | bubble | espresso (38), gown (4) |
| ConvNeXt | Euclidean | bf. ferret | M. hairless (21) |
| | | Saluki | schipperke (6) |
| | | schipperke | Saluki (6) |
| | | h.monkey | hippopotamus (5), water buffalo (4) |
| | | ambulance | beach wagon (3) |
| | | trombone | French horn (5) |
| | | cliff | slipper (10), flute (7) |
| | | artichoke | sea urchin (3) |
| | | bubble | espresso (3) |
| | Manhattan | trombone | French horn (4) |
| | Canberra | earthstar | sunscreen (6) |

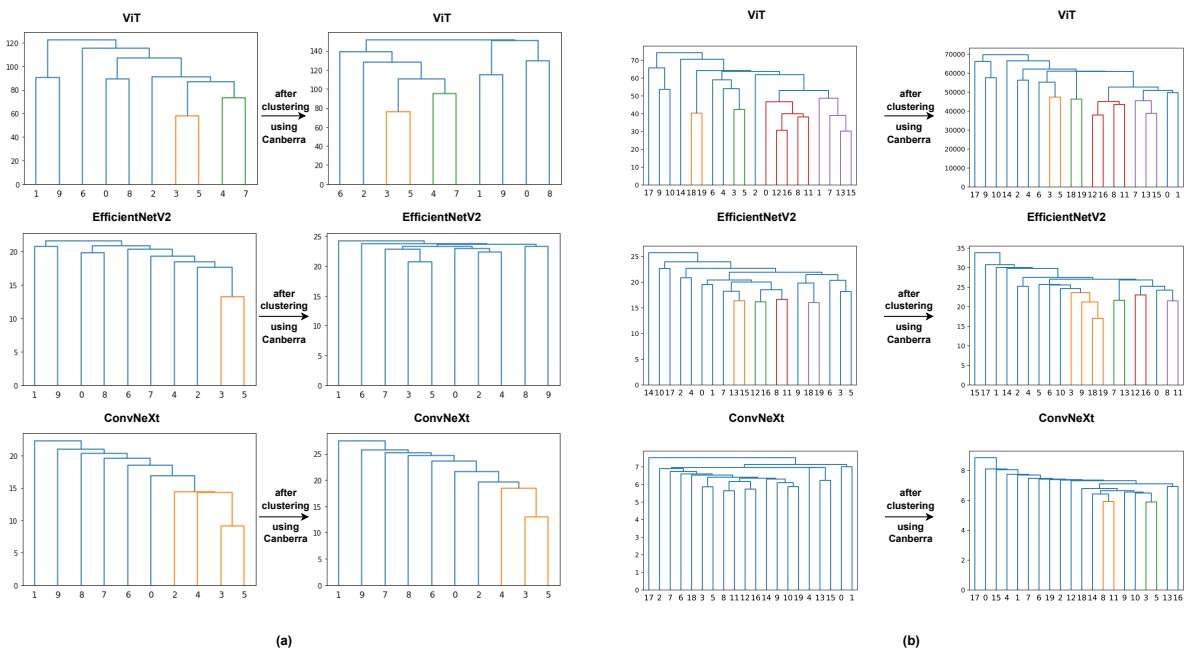

Figure 7: Dendrograms formed before and after clustering using Canberra distance. The UPGMA algorithm have used Euclidean distance to build the trees. The x-axis represents the class labels. (a) *CIFAR10* - (0) airplane, (1) automobile, (2) bird, (3) cat, (4) deer, (5) dog, (6) frog, (7) horse, (8) ship, (9) truck. (b) *CIFAR100(20)* (0) aquatic animals (1) fish (2) flowers (3) food containers (4) fruits and vegetables (5) household electrical vehicles (6) household furniture (7) insects (8) large carnivores (9) large man-made outdoor things (10) large natural outdoor scenes (11) large omnivores and herbivores (12) medium-sized mammals (13) non-insect invertebrates (14) people (15) reptiles (16) small mammals (17) trees (18) vehicles 1 (19) vehicles 2

where, $Dc(\Lambda_i)$ denotes clade deformation. Thus, deformity index computes the degree of deformation for each clade in the generated tree with respect to the reference tree. When the target tree is consistent with the reference tree, deformity index becomes 0. The maximum value is achieved when the tree is a caterpillar tree with the members of reference clades present at the highest levels. We use this measure to compare our relation tree with the *WordNet* hierarchy tree in Table 18.

The tree is built using three different measures, Euclidean, Manhattan and Canberra. The first column of Table 18 represent the models from which the features have been obtained. The second column denotes the distance metric used to form the feature representation cluster. The three columns under each datasets are measures used while forming the relation tree. From Table 18, we observe best results for *ViT* using Canberra distance measures when the tree is built using Euclidean distance on *CIFAR10* dataset. A minimum value of 0 has been observed which indicates no deformation with respect to the reference tree. However, in case of *EfficientNetV2*, under similar cluster measure, tree built using Canberra distance show minimum deformity for both *CIFAR100(20)* and *CIFAR100* datasets. *ImageNet* show best results with *EfficientNetV2* model. The overall deformity observed is least when a combination of *EfficientNetV2* model and Manhattan or Canberra measures are used for generating embeddings and clustering, respectively. While comparing the deformity index using cluster centroid, $G$, and manifold mean point, $P$, we observe better average results for the latter. In case of *CIFAR10* and *CIFAR100*, similar results are observed in most of the cases, however, in *ImageNet*, we observe substantially better results using $P$.

We summarise our observations as follows:

- Due to the manifold structure of the embeddings in the latent space, each cluster can be represented using $2n+4$ parameters such as, centroid $G$, manifold mean point $P$, maximum radius of the cluster,

Table 17: Results for Robinson-Foulds distance metric between the trees generated using UPGMA with centroid *G* /*MPP P* and *WordNet* hierarchy.
Left: *CIFAR10*, Right: *CIFAR100(20)*, Bottom Left: *CIFAR100*, Bottom Right: *ImageNet*

| Model | Cluster Distance fn. | Euclidean | | Manhattan | | Canberra | |
|---|---|---|---|---|---|---|---|
| | | G | MPP | G | MPP | G | MPP |
| *ViT* | Euclidean | 6 | 3 | 7 | 3 | 7 | 7 |
| | Manhattan | 6 | 7 | 7 | 7 | 7 | 7 |
| | Canberra | **1** | **1** | 3 | 3 | 5 | 5 |
| *EfficientNetV2* | Euclidean | 13 | 13 | 9 | 5 | 11 | 11 |
| | Manhattan | 13 | 13 | 9 | 9 | 10 | 11 |
| | Canberra | 13 | 13 | 5 | 5 | 10 | 11 |
| *ConvNeXt* | Euclidean | 13 | 13 | 13 | 13 | 13 | 13 |
| | Manhattan | 13 | 13 | 13 | 13 | 13 | 13 |
| | Canberra | 13 | 13 | 13 | 13 | 13 | 13 |
| *CLIP* | Euclidean | 9 | 5 | 9 | 5 | 13 | 5 |
| | Manhattan | 11 | 5 | 11 | 5 | 13 | 5 |
| | Canberra | 13 | 5 | 13 | 5 | 13 | 5 |

| Model | Cluster Distance fn. | Euclidean | | Manhattan | | Canberra | |
|---|---|---|---|---|---|---|---|
| | | G | MPP | G | MPP | G | MPP |
| *ViT* | Euclidean | 30 | 30 | 30 | 30 | 30 | 30 |
| | Manhattan | 30 | 30 | 30 | 30 | 30 | 30 |
| | Canberra | 28 | 26 | 28 | 26 | 30 | 26 |
| *EfficientNetV2* | Euclidean | 20 | 22 | 18 | 20 | 18 | 20 |
| | Manhattan | 22 | 24 | 16 | 18 | 18 | 20 |
| | Canberra | 20 | 22 | 16 | 18 | 18 | **14** |
| *ConvNeXt* | Euclidean | 24 | 24 | 24 | 22 | 16 | 18 |
| | Manhattan | 24 | 26 | 24 | 24 | 14 | 16 |
| | Canberra | 26 | 28 | 24 | 26 | 24 | 26 |
| *CLIP* | Euclidean | 28 | 16 | 28 | 14 | 22 | 18 |
| | Manhattan | 32 | 18 | 32 | 18 | 30 | 16 |
| | Canberra | 32 | 26 | 32 | 24 | 30 | 22 |

| Model | Cluster Distance fn. | Euclidean | | Manhattan | | Canberra | |
|---|---|---|---|---|---|---|---|
| | | G | MPP | G | MPP | G | MPP |
| *ViT* | Euclidean | 149 | 145 | 149 | 145 | 149 | 145 |
| | Manhattan | 149 | 149 | 149 | 149 | 149 | 147 |
| | Canberra | 143 | 141 | 139 | 135 | 145 | 133 |
| *EfficientNetV2* | Euclidean | **125** | 127 | 127 | **125** | 141 | 135 |
| | Manhattan | **125** | **125** | 127 | 127 | 135 | 135 |
| | Canberra | 135 | 129 | 139 | 131 | 139 | 137 |
| *ConvNeXt* | Euclidean | 135 | 143 | 143 | 143 | 129 | 133 |
| | Manhattan | 145 | 139 | 141 | 137 | 129 | 129 |
| | Canberra | 137 | 137 | 141 | 141 | 135 | 135 |
| *CLIP* | Euclidean | 147 | 129 | 147 | 129 | 147 | 129 |
| | Manhattan | 141 | 127 | 141 | 127 | 141 | 127 |
| | Canberra | 149 | 129 | 149 | 129 | 149 | 129 |

| Model | Cluster Distance fn. | Euclidean | | Manhattan | | Canberra | |
|---|---|---|---|---|---|---|---|
| | | G | MPP | G | MPP | G | MPP |
| *ViT* | Euclidean | 1365 | 1369 | 1365 | 1369 | 1365 | 1365 |
| | Manhattan | 1367 | 1369 | 1367 | 1369 | 1367 | 1369 |
| | Canberra | 1367 | 1369 | 1367 | 1367 | 1367 | 1369 |
| *EfficientNetV2* | Euclidean | 1361 | 1357 | 1359 | 1357 | 1361 | 1365 |
| | Manhattan | 1361 | **1355** | 1363 | 1357 | 1361 | 1361 |
| | Canberra | 1363 | 1359 | 1365 | 1359 | 1363 | 1361 |
| *ConvNeXt* | Euclidean | 1361 | 1361 | 1359 | 1359 | 1359 | 1357 |
| | Manhattan | 1361 | 1359 | 1361 | 1359 | 1361 | 1359 |
| | Canberra | 1363 | 1359 | 1361 | 1359 | 1361 | 1359 |
| *CLIP* | Euclidean | 1363 | 1363 | 1361 | 1363 | 1359 | 1363 |
| | Manhattan | 1369 | 1359 | 1369 | 1359 | 1367 | 1363 |
| | Canberra | 1363 | 1363 | 1365 | 1365 | 1367 | 1367 |

Table 18: Results for Deformity Index between the trees generated using UPGMA with centroid *G*/*MPP P* and *WordNet* hierarchy.
Left: *CIFAR10*, Right: *CIFAR100(20)*, Bottom Left: *CIFAR100*, Bottom Right: *ImageNet*

| Model | Cluster Distance fn. | Euclidean | | Manhattan | | Canberra | |
|---|---|---|---|---|---|---|---|
| | | G | MPP | G | MPP | G | MPP |
| *ViT* | Euclidean | 1.80 | 1 | 1.97 | 1 | 2.35 | 1.89 |
| | Manhattan | 1.80 | 1.46 | 1.80 | 1.46 | 2.35 | 1.89 |
| | Canberra | **0** | **0** | 1.50 | 1 | 1.90 | 1.89 |
| *EfficientNetV2* | Euclidean | 5.27 | 5.27 | 2.11 | 1.29 | 5.52 | 4.67 |
| | Manhattan | 4.71 | 4.71 | 1.87 | 2.11 | 3.77 | 4.55 |
| | Canberra | 3.78 | 3.77 | 0.83 | 1 | 4.89 | 5.35 |
| *ConvNeXt* | Euclidean | 5.47 | 5.46 | 5.28 | 5.28 | 4.84 | 4.70 |
| | Manhattan | 5.47 | 5.46 | 5.28 | 4.56 | 4.92 | 4.70 |
| | Canberra | 5.70 | 5.70 | 5.70 | 5.70 | 4.84 | 4.83 |
| *CLIP* | Euclidean | 4.74 | 1.16 | 4.74 | 1.16 | 2.41 | 1.25 |
| | Manhattan | 3.76 | 1.25 | 3.59 | 1.16 | 3.19 | 1.25 |
| | Canberra | 5.01 | 1.16 | 5.01 | 1.16 | 3.95 | 1.25 |

| Model | Cluster Distance fn. | Euclidean | | Manhattan | | Canberra | |
|---|---|---|---|---|---|---|---|
| | | G | MPP | G | MPP | G | MPP |
| *ViT* | Euclidean | 14.78 | 16.77 | 14.91 | 16.90 | 14.70 | 80.68 |
| | Manhattan | 13.69 | 18.23 | 15.95 | 18.16 | 15.23 | 15.43 |
| | Canberra | 11.12 | 11.19 | 12.93 | 10.33 | 12.79 | 8.50 |
| *EfficientNetV2* | Euclidean | 4.15 | 4.15 | 3.93 | 3.93 | 5.74 | 5.74 |
| | Manhattan | 8.83 | 8.83 | 3.03 | 3.03 | 5.74 | 5.74 |
| | Canberra | 7.65 | 7.65 | 5.90 | 5.90 | **2.97** | **2.97** |
| *ConvNeXt* | Euclidean | 10.66 | 13.24 | 10.44 | 11.41 | 8.61 | 8.41 |
| | Manhattan | 10.43 | 14.44 | 10.51 | 13.31 | 9.44 | 9.04 |
| | Canberra | 16.83 | 16.83 | 17.87 | 17.87 | 8.67 | 8.67 |
| *CLIP* | Euclidean | 10.54 | 3.56 | 10.52 | 3.80 | 8.36 | 4.56 |
| | Manhattan | 16.09 | 4.45 | 16.51 | 3.41 | 11.74 | 3.94 |
| | Canberra | 21.28 | 11.21 | 21.28 | 9.51 | 10.60 | 5.03 |

| Model | Cluster Distance fn. | Euclidean | | Manhattan | | Canberra | |
|---|---|---|---|---|---|---|---|
| | | G | MPP | G | MPP | G | MPP |
| *ViT* | Euclidean | 106.18 | 80.68 | 106.60 | 86.95 | 65.73 | 74.08 |
| | Manhattan | 92.63 | 104.10 | 100.33 | 111.69 | 68.49 | 80.36 |
| | Canberra | 51.25 | 66.59 | 45.88 | 65.68 | 55.41 | 40.36 |
| *EfficientNetV2* | Euclidean | 18.57 | 17.16 | 14.61 | 12.16 | 24.85 | 23.91 |
| | Manhattan | 18.73 | 18.73 | 13.61 | 13.61 | 23.77 | 23.77 |
| | Canberra | 63.48 | 18.13 | 43.22 | **11.99** | 29.15 | 28.52 |
| *ConvNeXt* | Euclidean | 96.45 | 77.10 | 90.56 | 66.64 | 21.38 | 23.10 |
| | Manhattan | 112.70 | 93.43 | 104.61 | 89.16 | 25.16 | 24.34 |
| | Canberra | 84.19 | 84.19 | 94.07 | 94.07 | 28.46 | 28.46 |
| *CLIP* | Euclidean | 111.57 | 17.73 | 114.01 | 17.94 | 30.07 | 19.23 |
| | Manhattan | 134.20 | 21.83 | 137.31 | 21.53 | 34.02 | 16.05 |
| | Canberra | 150.46 | 21.58 | 154.75 | 22.99 | 37.21 | 17.13 |

| Model | Cluster Distance fn. | Euclidean | | Manhattan | | Canberra | |
|---|---|---|---|---|---|---|---|
| | | G | MPP | G | MPP | G | MPP |
| *ViT* | Euclidean | 2547.64 | 2141.26 | 2583.87 | 2122.93 | 585.37 | 810.35 |
| | Manhattan | 2592.95 | 2239.78 | 2522.28 | 2186.02 | 400.11 | 635.56 |
| | Canberra | 2005.55 | 614.38 | 2013.02 | 632.08 | 1220.78 | 404.76 |
| *EfficientNetV2* | Euclidean | 1031.02 | 292.65 | 135.82 | **125.78** | 177.88 | 154.75 |
| | Manhattan | 1393.86 | 290.07 | 157.72 | 127.95 | 179.68 | 162.67 |
| | Canberra | 1869.49 | 272.96 | 202.24 | 148.03 | 241.37 | 157.98 |
| *ConvNeXt* | Euclidean | 546.33 | 399.16 | 821.04 | 464.64 | 174.91 | 142.69 |
| | Manhattan | 476.94 | 427.40 | 1231.60 | 483.76 | 196.92 | 135.76 |
| | Canberra | 466.12 | 614.15 | 1113.68 | 615.80 | 228.68 | 125.45 |
| *CLIP* | Euclidean | 176.34 | 156.37 | 202.72 | 181.49 | 164.97 | 140.79 |
| | Manhattan | 189.26 | 167.94 | 230.12 | 207.75 | 149.76 | 131.56 |
| | Canberra | 206.04 | 184.11 | 242.72 | 208.14 | 150.68 | 133.97 |

minimum radius of the cluster, the angle $\frac{\theta_{max}}{2}$ between the mean vector $\vec{\mu}$ and vector passing through the rim of the cluster surface, and the angle $\frac{\theta_{min}}{2}$ between the mean vector $\vec{\mu}$ and vector passing through the central annular ring.

- The average result using Canberra measure is better when all the metrices are compiled.

- The embeddings formed in the space form well defined regions with minimum overlap with other classes in case of *EfficientNetV2* and *ConvNeXt*. However, in case of *ViT*, multiple overlaps with unrelated classes are observed.

- *EfficientNetV2* is able to capture the semantic relationship among the classes present in the dataset. However, it is not distinct in *ViT*.

- For high dimensional data, cluster semantics for all three distance measures behave similarly.

- The Robinson-Foulds distance and Deformity Index results observed using $P$ are either equal or better compared to $G$ while measuring the similarity between the relation trees and *WordNet*. When the number of classes are less, for example, *CIFAR10*, *CIFAR100(20)* and *CIFAR100*, the score is similar for centroid $G$ and MPP $P$. However, in case of *ImageNet*, we observe substantial improvement when $P$ is used.

### 4.6 Hierarchy among label embeddings

We compare the hierarchical relations formed using word embeddings of the image labels from two pre-trained models, namely, *BERT* (Devlin et al., 2018) and *GloVe* (Pennington et al., 2014) with the given *WordNet* ontology. In this experiment, we form captions for each of the images present in *CIFAR10*, *CIFAR100(20)*, *CIFAR100* and *ImageNet* datasets using the labels. The word embeddings generated from the pre-trained models are used to form relation trees by applying UPGMA algorithm using Euclidean, Manhattan and Canberra distance metrics. We compare the relation trees with a sub-tree derived from *WordNet* using Robinson-Foulds and Deformity Index in Table 19. From Table 19, we observe that the hierarchical

Table 19: Results for Robinson-Foulds distance (RF) and Deformity Index (DI) between the trees generated using UPGMA with label embeddings and *WordNet* hierarchy.

| Model | UPGMA Distance fn. | CIFAR10 | | CIFAR100(20) | | CIFAR10 | | ImageNet | |
|---|---|---|---|---|---|---|---|---|---|
| | | RF | DI | RF | DI | RF | DI | RF | DI |
| *BERT* | Euclidean | 15 | 6.36 | 28 | 8.83 | 143 | 34.03 | 1367 | 299.64 |
| | Manhattan | 15 | 6.36 | 28 | 8.83 | 143 | 36.19 | 1367 | 296.39 |
| | Canberra | 15 | 5.55 | 28 | 7.15 | 139 | 26.94 | 1365 | 174.82 |
| *GloVe* | Euclidean | 15 | 7.07 | 30 | 18.69 | 149 | 143.84 | 1365 | - |
| | Manhattan | 15 | 7.07 | 30 | 18.69 | 149 | 143.84 | 1365 | - |
| | Canberra | 15 | 7.07 | 30 | 18.69 | 149 | 143.84 | 1365 | - |

relationships captured by the image embeddings are more meaningful compared to the hierarchy formed by its corresponding captions. However, if we compare the two language models, *BERT* performs better than *GloVe* when the number of classes are increased.

## 5 Conclusions

In this paper, we have analysed the structure of the embedding space when different types of models are used. Moreover, we have been able to establish a hierarchical relationship among interacting classes using relation trees. These trees have been evaluated using phylogenetic tree comparison methods. Further, we have proposed a cluster growing technique to minimise the overlap and inclusion of other classes to form high quality clusters of embeddings. A comparative study among the different distance measures used for

clustering has shown that Canberra distance outperforms the other measures to form better quality clusters with minimum overlaps and maximum coverage. From our experiments we observe that *EfficienNetV2* show minimum interaction among the classes while training, and form non-overlapping clusters using our technique. The mean vector computation is more robust and has shown better results compared to centroid while comparing the hierarchical trees. However, we have not considered the involvement of the $\theta$ parameter while computing the distance between any two clusters. A method to include both $\theta$ and mean vector for tree comparison technique will be more appropriate and robust.

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

## A Chi-squared test setup

The chi-squared test is conducted for all the datasets to show that the distribution followed by the embeddings is Poisson in nature. We divide the distance between minimum and maximum radius into 22 intervals. Each of these intervals cover a radius of 5 units. We tabulate the number of embeddings from target and non-target classes for each interval for a given cluster. We perform the chi-squared test to compare this distribution with the Poisson distribution of similar mean and standard deviation. The results are shown in Section 3.2.1. This experiment has been conducted on randomly selected 10 clusters from each dataset. The results shown in Section 3.2.1 is for *CIFAR10* dataset.

## B Dendrograms for *CIFAR100* datasets

We have shown the dendrograms of *CIFAR10* and *CIFAR100(20)* datasets after clustering in Fig. 7. In this section, we show the results for *CIFAR100* dataset on *EfficientNetV2* and *ConvNeXt* in Fig. 8.

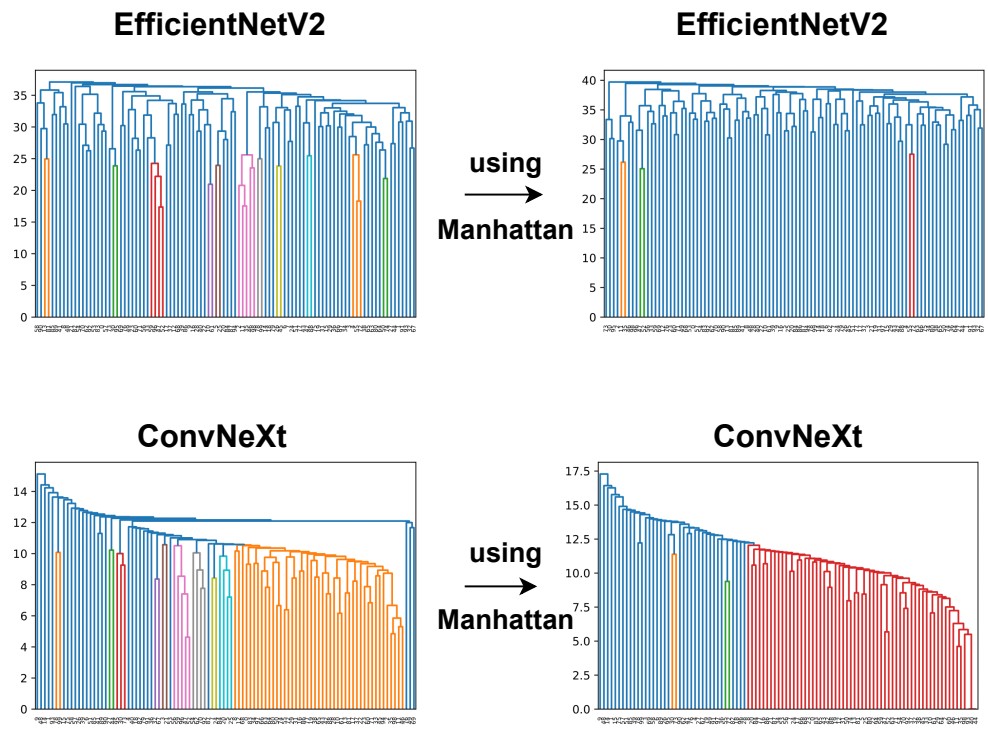

Figure 8: Dendrograms formed before and after clustering using Manhattan distance. The UPGMA algorithm have used Euclidean distance to build the trees.

