# OpenReview forum: "Discovery of Hierarchy in Embedding Space"
_TMLR — Rejected by TMLR_

### Review · Reviewer_T8qQ · 2023-11-27

**Summary Of Contributions:**

The paper proposes an algorithm for growing clusters in the embedding space of a vision model, aiming to minimize the inclusion of other classes and form clusters of similar representations​. The resulting clusters can be overlapped to represent ambiguous embeddings that cannot be mapped to a particular class with high confidence. The paper evaluates the quality of the clustering and compares the hierarchical relation trees formed, using phylogenetic tree comparison methods to assess the method against the WordNet hierarchy.

**Audience:**

Yes

**Claims And Evidence:**

No

**Requested Changes:**

1. Include a detailed comparison with other clustering and supervised learning methods, highlighting the distinctions and advantages of the proposed method.
2. Integrate a more comprehensive review of related works, especially focusing on visualization and manifold learning methods.
3. Address the typographical and grammatical errors, and provide clearer explanations about the training methodologies of the considered vision models. Also, structure the presentation of findings in a more coherent and distinct manner, separating theoretical and empirical results.
4. Broaden the scope of the experiments to include other data types (e.g. NLP) beyond image datasets, ensuring the generalizability of the proposed method.

**Strengths And Weaknesses:**

#### Strengths

- The paper makes a step forward towards an in-depth analysis of the embedding spaces generated from different vision models.
- The paper thoroughly evaluates the clusters' structures and relationships, offering insights into the latent space's structures.
- The methodology is tested across various models (ViT, EfficientNetV2, ConvNeXt), showcasing its adaptability and robustness.

#### Weaknesses
- The proposed method, while positioned in the realm of clustering, relies on class information. This makes the positioning of the method ambiguous.
- The proposed method relies on class information, raising questions about its comparison to supervised learning methods such as k-NN classifiers. Moreover, it raises question about its comparison to the Gaussian mixture model, which also allows overlapping? A more detailed discussion on these aspects would be highly beneficial.
- The paper does not sufficiently cover related works, especially in areas like visualization methods and manifold learning methods, which are integral to this field of study.
- The paper suffers from several presentation issues, which largely limits its readability. Specifically,
	- There are numerous typographical and grammatical errors (see below).
	- Absence of detailed explanations regarding the training and fine-tuning of vision models such as ViT, EfficientNetV2, and ConvNeXt. No context and experimental setup are given.
	- Theoretical and empirical findings are intermixed in a confusing manner, particularly in Section 3.2, where empirical proofs lack clarity and detailed experimental setup descriptions. Therefore, It is very hard to understand the soundness and relevance of each statement.
	- In Section 3.2.2, the significance of calculated values like $\theta_{\min}$ and $\theta_{\max}$ for various datasets is not discussed, leaving a gap in the understanding of their relevance.
- The experiments focus primarily on image datasets (CIFAR10, CIFAR100, ImageNet), which may limit the generalizability of the findings to other types of data​

#### Identified Typos and Grammatical errors

- Inconsistent capitalization:
	- "vision transformers" vs. "Vision Transformers" (Page 3)
	- "EfficientNetV2" vs. "Efficient Net V2" (Page 3)
- Grammatical errors
	- "The embeddings generated by these models using our cluster growing and quality analysis algorithm." (The sentence structure seems incomplete)
	- Page 4: "due to its..." -> "due to their"
	- Page 8: "The p-value generated for this experiment if 0.00008"?
	- Last page: "ha sbeen" -> "has been"

---

> ### Author Response · Authors · 2024-01-04
> **Response to Reviewer T8qQ**
>
> Prior labels: We analyse an already encoded space using a cluster growing technique where the interaction among various classes are studied. Class information is used only while detecting the centroids. However, the overall importance of cluster growing technique is to analyse the quality of embeddings and deriving the hierarchical relationships inherently reflected in the latent space without using any external information while training. The proposed method does not modify the data points present in the embedding space, rather it selects the points to be included or discarded to form well defined regions representing discriminating, ambiguous and rejected embeddings to depict a particular class.
>
> KNN and GMM- The proposed method is similar to K-NN classifiers where instead of including “K” nearest embeddings and categorising them into the majority class, we include all the embeddings present within a particular radius and use the original class information to analyse how embeddings of other classes intrude. This enables us to find how similar and dissimilar classes impact the performance of various models which leads to a drop in the overall accuracy of the network. Although the Gaussian mixture model allows overlapped clusters, it assumes that the data points in each cluster follow Gaussian distribution. However, empirically we have observed that in the latent space, it mostly does not follow Gaussian distribution. Poisson distribution is a more probable candidate in this regard, although some clusters may follow Gaussian.
>
> Related Works: In Section 2, we have included more related works covering visualization and manifold learning methods in the revised manuscript.
>
> Presentation issues: The typographical and grammatical errors have been corrected in the revised paper. We have used the pre-trained models for ImageNet dataset. However, for CIFAR10 and CIFAR100 datasets we have fine-tuned the pre-trained models for 50 epochs using stochastic gradient descent optimizer, keeping the learning rate fixed at 0.001. We have included the details in Section 4 of the revised manuscript. In Section 3.2.1, Theorem 1 and 2 are through empirical findings and Theorem 3 has been theoretically derived using the previous two empirical results. The description of the experimental setup for the empirical proofs have been included in Appendix A. The data points in a cluster lie between theta min and theta max computed with respect to the centroid. Thus, giving us an insight to the structure of a cluster in the embedding space. A detailed explanation on the significance of these parameters have been discussed in Section 3.2.2 of the revised paper.
>
> Image data: In this study, we explore the quality of image embedding and how different image based models behave differently on various datasets when closely related classes interact with each other which gets reflected in the latent space. Therefore, we observe difference in performance and misclassification. The work primarily focuses on the inherent hierarchical relationships captured in the embedding space by various models and a way to analyse them both qualitatively and quantitatively. In Section 4.6 of the revised paper, we have included experiments using language models on image captions.
>
> Requested changes: 1) The motivation of this work is to analyse the interaction of various classes in the embedding space and compare the behaviour of different types of models using the same dataset. The comparison has been made between the three types of models with respect to the image embeddings in the latent space. To quantify the inter-class interaction, we propose the cluster growing technique and generate the hierarchy formed in the encoded space. The hierarchy is compared with the existing WordNet ontology for different models. The proposal of the cluster growing technique is to qualitatively and quantitatively define the latent space with respect to the image embeddings.
>
> 2) We have included more review of related works in Section 2 of the revised paper.
>
> 3) The typographical and grammatical errors have been corrected in the revised paper.
>
> 4) Experiments using language models on image captions have been shown in Section 4.6 of the revised paper. The experiments include comparison of hierarchical relationship among the class label embeddings. We use BERT and GloVe to generate word embeddings for each image caption and form a relation tree using UPGMA algorithm and compare it with the WordNet ontology.

---

### Review · Reviewer_LgVY · 2023-12-09

**Summary Of Contributions:**

In this work the authors propose a method for identifying a soft clustering of embeddings. In particular, this soft clustering starts by identifying a centroid $G_k$ for each class $k \in \mathbb C$. Then, for each class $k$, a maximum radius $r_b$ is decided such that the proportion of embeddings within $r_b$ of $G_k$ which are not of class $k$ is sufficiently small (less than $\gamma$, a hyperparameter). The eventual radius, $r_f$, is chosen to be the radius less than $r_b$ such that increasing this radius by $1$ would yield the largest increase in the number of elements of class $k$ contained in this radius. This defines the cluster. The authors go on to analyze these clusters, as well as the implied relationships between these clusters, using a variety of image embedding models and distance metrics.

**Audience:**

No

**Claims And Evidence:**

No

**Requested Changes:**

1. As the centers are defined using the centroid of the embeddings corresponding to a particular label, it seems the main methodological contribution of the paper is to propose a particular radius, however there is no real justification for this selection of the radius. Can you provide any justification for this choice of radius? Overall, what is the main contribution of this work?

2. Assumption 1 states: "We assume that each of these clusters form a hypersphere in the latent space." What does this mean? The cluster is a finite set of points in $\mathbb R^d$, and a hypersphere is a $d-1$ dimensional manifold. The text which follows Assumption 1 seems to suggest that the intention was to describe that all the points in the cluster can be contained within a hypersphere, with one of the points on the boundary, but this is clear from the way the clusters are defined and does not require an assumption. There is a similar problem with Assumption 2, in that the authors attempt to validate their assumption, which is not required of an assumption.

3. All theorems in this work are false. For theorem 1, for example, one can easily create a counter-example with 1 class in 2 dimensions, by placing $(n % 2) * n$ embeddings at radii $n$ away from a centroid. The support for theorem 2 relies on some empirical evidence, but that is not a proof. Theorem 3 is missing some explicit dependence on assumptions, and also relies on Theorem 2, which was not actually proved but just supported by some empirical evidence. Overall, the writing and structure of section 3.2.1 requires a significant rewrite with a focus on logic and correctness.

4. The reason to use a fixed increase of 1 unit in the radius seems odd, as does the final selection of $r_f$ described in equation (4). Why would we want to stop when the number of target class embeddings increases the most? Is there a principled reason for using the absolute difference as opposed to, say, a relative difference?

2. In equation (7), the mean vector $\vec \mu_k$ is defined as
$$\vec \mu_k = \frac{\sum_{i=1}^n e_{ik} - G_k}{n}.$$
If this definition is correct as written, then this seems to be equivalent to $\frac{n-1}n G_k$. If the $G_k$ was intended to be contained within the summation, then this would be $0$. What is the correct definition, and what is the significance of $\vec \mu_k$?

3. What is the significance of the manifold mean point $P = G_k + \vec \mu_k$, and the relative angles presented in table 2?

4. There seems to be a typo in equation (17), the sum is taken over $i$ but there is no $i$ in the summand.

**Strengths And Weaknesses:**

The authors have presented extensive analysis and tables of results in the main work, and have made an effort to depict the various aspects of their proposed algorithm.

Unfortunately, this paper would require work significant before it would be suitable for acceptance in a conference or journal. There are two main weaknesses: (1) it is unclear to me what benefit the proposed algorithm provides, and (2) all theorems contained within the paper are false.

---

> ### Author Response · Authors · 2024-01-04
> **Response to Reviewer LgVY**
>
> Requested changes: 1) In all our experiments on CIFAR10, CIFAR100 and ImageNet datasets, we have observed that the volume covering the centroid region does not contain any image embedding. The first embedding of any cluster occurs only after a minimum radius is covered. The boundary radius for each cluster has been computed based on our algorithm. Thus, the selection of both minimum and maximum radius is based on observation.
>
> 2) We are thankful to the reviewer for pointing this out. The assumptions are mostly definitions stated to empirically validate our observation. We have modified them in the revised paper.
>
> 3) We analyse an already encoded space using a cluster growing technique where the interaction among various classes are studied. The proposed method does not modify the data points present in the embedding space, rather it selects the points to be included or discarded to form well defined regions representing discriminating, ambiguous and rejected embeddings to depict a particular class.  In Section 3.2, the theorems are based on our observations while conducting various experiments on different datasets. Similar trends were observed for each of the datasets. Therefore, these theorems were stated along with empirical evidences.
>
> 4) The algorithm counts the number of embeddings from target and non-target class per unit radius. It stops when the maximum increase of target embeddings are included. This radius is denoted as rf. As we increase the radius beyond rf, the number of target embeddings reduce while the non-target embeddings start increasing rapidly. To avoid the inclusion of non-target embeddings, the algorithm stops at rf.
>
> 5) The centroid is intended to be inside the summation. The mean vector is calculated for the final cluster which does not contain all the embeddings of a target class while the centroid of the class is calculated considering all the embeddings present in a class. Therefore, the summation does not become 0.
>
> 6) When we visualize the clusters using t-SNE plots, we observe that the embeddings are contained together at a particular distance from the centroid as shown in Figure 4. The manifold mean point represents a resultant point within the volume containing all the embeddings. The angles are computed with respect to the structure formed by a cluster in the latent space. The embeddings are present within the minimum and maximum angle calculated from the centroid. All these parameters are calculated to define the structure as well as represent each cluster.
>
> 7) We are grateful to the reviewer for pointing out this mistake. It has been rectified in the revised paper.

---

> > ### Comment · Reviewer_LgVY · 2024-01-18
> >
> > I appreciate the response from the authors and the efforts to address the issues I proposed, however I still feel very strongly that the paper has very serious issues from a soundness perspective, and furthermore I am still at a loss as to what the main contribution of the paper is. After the authors clarification, I believe that the main contribution seems to be a "cluster growing method" which is proposed for, adapted to, and analyzed on a particular embedding model and dataset. As identified in my original review, many choices in the proposed algorithm seem arbitrary, and since the method is not compared with any baseline there is no empirical justification for these arbitrary choices.
> >
> > More concerning is that results which are dataset and model dependent are presented as "theorems". At best, these results are observations. For example, the statement of "Theorem 1" is written: "The distribution representing the number of embeddings occurring per unit radius, $|Y_{r+1}^n| - |Y_r^n|$, peaks at a particular radius and then starts decreasing." If this were to be interpreted as a *theorem* it is quite interesting, however (1) it is only true with respect to the model and dataset under investigation, and (2) even though this is the case, the authors seem to attempt to give a sort of "proof" which amounts to saying that the number of embeddings are finite, and therefore for large enough radius this value will be small, as will the case for small enough radius, and thus the maximum must occur somewhere. This is hardly worth calling a proof, because it just the observation that a bounded discrete function with compact support attains a maximum somewhere, and moreover does not prove the statement in question because (as the authors go on to note) there may be multiple peaks, and the authors simply mention "Empirically, we have found that there exists only one such peak per cluster".
> >
> > This particular case is indicative of the sort of arguments made throughout the work, and I only give such detail as to support my claim that this paper is fundamentally unsound. I would encourage the authors to more clearly identify the main purpose of their paper, and to revisit the sections with mathematical details as there continue to be substantial issues with correctness in each section I have looked at carefully.

---

> > > ### Author Response · Authors · 2024-01-19
> > > **Response to Reviewer LgVY**
> > >
> > > Our main contribution is studying the quality of embeddings and analysing the hierarchical relationships among them. The cluster growing technique only forms a method to define the boundaries of the classes which discriminate distinct embeddings from the ambiguous ones. The comparison has been made on different models and datasets to study their embeddings. Comparison with different techniques focus on the cluster growing method rather than the quality of the embeddings. Therefore, comparing clustering methods in this case becomes irrelevant.
> > >
> > > We appreciate the comments from the reviewer. The empirical observations may be treated as hypothesis rather than theorems. These trends have been followed by all the models and datasets that we have used. Hence, technically we cannot call them as theorems but may present them as hypothesis. The necessary modifications have been made in the revised manuscript. We have also included results from CLIP model in Table 17 & 18.

---

### Review · Reviewer_A5r5 · 2023-12-24

**Summary Of Contributions:**

In this work, authors studied the latent space structure of embeddings by proposing a cluster-growing algorithm based on maximizing dominating class aggregation while minimizing inclusion of other classes. The authors stated certain assumptions and properties of the cluster-growing algorithm and applied the method to several datasets with various backbone models.

**Audience:**

Yes

**Broader Impact Concerns:**

I don't have ethical concerns about this work.

**Claims And Evidence:**

No

**Requested Changes:**

1. I wonder if authors could provide some insights on the cases when some assumptions are violated, for example, when true class labels are unknown (which is usually true for clustering setting), and when classes are not balanced (there may be dominating class with a majority of the data points), is it still beneficial to apply this method?
2. More experiments comparing the proposed method and existed work should be added. Current experiment results need to be refined as it only compares a limited range of models and image datasets.
3. For the theoretical analysis, I have following confusions and would appreciate if authors can address them:
    1. For section 3.2.1 Assumption 1, I am curious if the authors could elaborate more on how often this assumption holds for the embedding models.
    2. For section 3.2.1 Theorem 1, is this purely from empirical result, if so, what data (dataset, embedding model, class selected as target class) is used for the experiment? What is the ‘particular’ radius?
    3. For section 3.2.1 Theorem 2, authors used Chi-square test to prove increased number of embeddings per unit radius follows a Poisson distribution. However, p-val>0.05 only suggests there is no statistically significant evidence to reject null hypothesis, meaning we can’t reject the possibility that the data follows Poisson distribution, but neither can we conclude the data is from Poisson distribution.
    4. For section 3.2.1 Assumption 2, what is ‘a particular interval of radius‘ ?
    5. For section 3.2.3, I am curious why Euclidean distance is considered while in Assumption 1 the embeddings are assumed to be a hypersphere.
4. Settings are unclear in a lot of sections, as I pointed out in section 3.2 it is unclear how theorems are justified by the empirical results. Are assumptions 1, 2, 3 required for experiments or mainly for theoretical results, how realistic are they?
5. Related work and literature are too general and not closely related to this work, specifically discovering the hierarchy structure in the embedding space.

**Strengths And Weaknesses:**

**Strength:**
1. The idea that semantically related classes may interact with each other in the latent space is natural.

**Weaknesses:**
1. The assumptions and statements made by the authors are not convincing that the proposed method can be applied to real applications and aren’t enough to justify their claimed contributions.
2. The proposed cluster-growing technique assume the prior knowledge of true classes labels, which is uncommon and needs more justifications.
3. Theoretical results are not formed rigorously and are not convincing.
4. Empirically, this work does not have enough comparison with other existed/related works.

---

> ### Author Response · Authors · 2024-01-04
> **Response to Reviewer A5r5**
>
> Application-The proposed method puts forward five parameters to represent each cluster. We use these parameters to define class representations. The mean vectors computed for each of these classes along with the minimum and maximum angles and radii form a specific structure of a class which can be further applied in continual learning. When similar classes are encountered, the embeddings formed will lie within the structure of the class while for new classes, separate structure representations will be formed in the latent space. This application requires a series of experiments and will be explored as a future work.
>
> Prior labels-We analyse an already encoded space using a cluster growing technique where the interaction among various classes are studied. Class information is used only while detecting the centroids. The overall importance of cluster growing technique is to analyse the quality of embeddings and deriving the hierarchical relationships inherently reflected in the latent space without using any external information while training. The method only selects the points to be included or discarded to form well defined regions representing discriminating, ambiguous and rejected embeddings to depict a particular class.
>
> Comparison - The motivation of this work is to analyse the interaction of various classes in the embedding space and compare the behaviour of different types of models using the same dataset. The comparison has been made between the three types of models with respect to the image embeddings in the latent space. To quantify the inter-class interaction, we propose the cluster growing technique and generate the hierarchy formed in the encoded space. The hierarchy is compared with the existing WordNet ontology for different models. The proposal of the cluster growing technique is to qualitatively and quantitatively define the latent space with respect to the image embeddings.
>
> Requested changes-1) The mean vectors, angles and radii form class representations for each of the labels present. When true class labels are unknown, the vector may not belong to any of the clusters corresponding to classes. In that case, it indicates a new class and similar embedding may be grouped under the same new class. Handling class imbalance is a challenge on learning representative embeddings of a class. The analysis throws insight on how the minority class is separated from the majority class.
>
> 2) In our revised manuscript, we have included language models and shown results on word embeddings of image captions in Section 4.6. The experiments include comparison of hierarchical relationship among the class label embeddings. We use BERT and GloVe to generate word embeddings for each image caption and form a relation tree using UPGMA algorithm and compare it with the WordNet ontology.
>
> 3)1) In all our experiments on CIFAR10, CIFAR100 and ImageNet datasets, we have observed that the volume covering the centroid region does not contain any image embedding. The first embedding of any cluster occurs only after a minimum radius is covered. The boundary radius for each cluster has been computed based on our algorithm. Maximum number of embeddings occur between the minimum and boundary radius following two probable distributions, namely, Poisson or Gaussian distribution.
>
> 3)2) Theorem 1 has been presented based on empirical observations. The experiments have been conducted on CIFAR10, CIFAR100 and ImageNet datasets on Vision transformer, EfficientNetV2 and ConvNeXt models. In particular, the theorems in section 3.2.1 are presented based on empirical results on CIFAR10 dataset on Vision transformer and EfficienNetV2 model. The target class include the classes of CIFAR10. Similar trends are observed for other datasets as well.
>
> 3)3) We are thankful to the reviewer for pointing this out. The Chi-square test was conducted on other distributions as well from which we concluded that most of the clusters may follow Poisson distribution as we could not reject the null hypothesis. However, for a few clusters p-value>0.05 has been observed for Gaussian distribution as well. Therefore, we cannot reject the possibility that the clusters follow either Poisson or Gaussian distribution.
>
> 3)4) The interval considered in Assumption 2 is 5 units of radius.
>
> 3)5) Section 3.2.3 describes the distance measures used during the cluster growing algorithm. We use Euclidean distance to measure the distance between two image embeddings while forming the clusters. In Assumption 1, we assume each cluster to represent a hypersphere.
>
> 4) In Section 3.2, the theorems are based on our observations while conducting various experiments. Theorem 1 and 2 are justified using empirical results while Theorem 3 is a mathematical proof. The assumptions are required during the empirical proofs and not while executing the experiments.
>
> 5) In Section 2, we have included some related work on manifold learning methods and hierarchy structures in embedding space.

---

> > ### Comment · Reviewer_A5r5 · 2024-01-20
> >
> > I appreciate the effort authors have made to address my concerns. However, there are still obvious flaws of this work that need to be improved before being published. As reviewer LgVY pointed out, selections in this paper seems to be randomly made and not well-justified. As I first pointed out in my review, there are comparison between the proposed method and existed literature, Author argued in the response that the results are conducted across several different backbone models and compared with 'ground-truth' hierarchy, is not convincing to me the proposed method is valuable enough. Also, most of the discoveries are purely empirical and not general enough for different settings. Moreover, there are flaws in details that need careful examination. For example, although authors addressed my requested change 3.3 for hypothesis testing by adding one more null hypothesis that data can be from Guassian distribution as well. This method is simply misused in this case and it's not leading to the conclusion that data is from Guassian or Poisson distribution.

---

> > > ### Author Response · Authors · 2024-01-22
> > > **Response to Reviewer A5r5**
> > >
> > > We appreciate the comments made by the reviewer. However, our main contribution seems to have gone unnoticed. As the title of our paper suggests, “Discovery of Hierarchy in Embedding Space”, we study the semantic relationship among the classes formed in the embedding space and quantify the quality of embeddings. We are not proposing a clustering method in this paper. The cluster growing technique is formulated only to study the class embeddings, its formation and its hierarchical relationships in the latent space. Therefore, the comparison has been made across different models, such as ConvNeXt, EfficientNetV2, ViT and CLIP on CIFAR10, CIFAR100 (both broad and fine) and ImageNet with various distance metric (Euclidean, Manhattan and Canberra) rather than clustering techniques. Similar study on label embeddings using BERT and GloVE has also been presented as an additional study. We are thankful to the reviewer for pointing out our misjudgment in the hypothesis testing after which we carefully verified with all the distributions. We modified our statements by not ascertaining that the data is from Gaussian or Poisson distribution. We have only mentioned that it may belong to these distributions as we cannot conclusively reject the null hypothesis. The results are empirical and we have shown this trend on various types of models. Till now we have not found a counter example to reject this claim.

---

### Decision · Action_Editor_aUzf · 2024-01-29

**Recommendation:** Reject

**Comment:**

This paper introduces a clustering approach that is meant as a type of post-hoc analysis tool. It obtains radii for a type of cluster meant to be semantically meaningful (but requires class information to obtain the centers, so works only as a post-supervised training approach). The authors analyze this approach in several ways.

However, there are two concerns here; these are noted by all reviewers.

First, the evidence for the analysis and claims made is relatively weak: there are no proofs, and only limited experimental support.

Second, it is not clear what the value or application of the proposed approach is.

Overall the paper is in a preliminary state and with substantial growth could produce interesting results, but it is not there yet.

**Audience:**

The paper studies a form of clustering, so fits one of the audience interest areas.

**Claims And Evidence:**

No; much of the paper's contributions are a series of hypotheses that do not have sufficient evidence for them. This assessment is shared by all reviewers. I also agree.